# Multi-species, multi-transcription factor binding highlights conserved control of tissue-specific biological pathways

Benoit Ballester[1,2,3]*[†], Alejandra Medina-Rivera[4], Dominic Schmidt[5‡], Mar Gonzàlez-Porta[1], Matthew Carlucci[4], Xiaoting Chen[6], Kyle Chessman[4], Andre J Faure[1], Alister PW Funnell[7], Angela Goncalves[1], Claudia Kutter[5], Margus Lukk[5], Suraj Menon[5], William M McLaren[1], Klara Stefflova[5§], Stephen Watt[5¶], Matthew T Weirauch[8,9], Merlin Crossley[7], John C Marioni[1], Duncan T Odom[5,10], Paul Flicek[1,10], Michael D Wilson[4,5,11]*[†]

[1]European Molecular Biology Laboratory, European Bioinformatics Institute, Wellcome Trust Genome Campus, Cambridge, United Kingdom; [2]Aix-Marseille Université, UMR1090 TAGC, Marseille, France; [3]INSERM, UMR1090 TAGC, Marseille, France; [4]Genetics and Genome Biology Program, SickKids Research Institute, Toronto, Canada; [5]Cancer Research UK–Cambridge Institute, University of Cambridge, Cambridge, United Kingdom; [6]School of Electronic and Computing Systems, University of Cincinnati, Cincinnati, United States; [7]School of Biotechnology and Biomolecular Sciences, University of New South Wales, Kensington, Australia; [8]Center for Autoimmune Genomics and Etiology, Cincinnati Children's Hospital Medical Center, Cincinnati, United States; [9]Divisions of Biomedical Informatics and Developmental Biology, Cincinnati Children's Hospital Medical Center, Cincinnati, United States; [10]Wellcome Trust Sanger Institute, Wellcome Trust Genome Campus, Cambridge, United Kingdom; [11]Department of Molecular Genetics, University of Toronto, Toronto, Canada

**\*For correspondence:** benoit.ballester@inserm.fr (BB); michael.wilson@sickkids.ca (MDW)

[†]These authors contributed equally to this work

**Present address:** ‡Syncona Partners LLP, London, United Kingdom; ¶Wellcome Trust Sanger Institute, Wellcome Trust Genome Campus, Cambridge, United Kingdom; §Division of Biology and Biological Engineering, California Institute of Technology, Pasadena, United States

**Competing interests:** The authors declare that no competing interests exist.

**Reviewing editor**: John Stamatoyannopoulos, University of Washington, United States

**Abstract** As exome sequencing gives way to genome sequencing, the need to interpret the function of regulatory DNA becomes increasingly important. To test whether evolutionary conservation of cis-regulatory modules (CRMs) gives insight into human gene regulation, we determined transcription factor (TF) binding locations of four liver-essential TFs in liver tissue from human, macaque, mouse, rat, and dog. Approximately, two thirds of the TF-bound regions fell into CRMs. Less than half of the human CRMs were found as a CRM in the orthologous region of a second species. Shared CRMs were associated with liver pathways and disease loci identified by genome-wide association studies. Recurrent rare human disease causing mutations at the promoters of several blood coagulation and lipid metabolism genes were also identified within CRMs shared in multiple species. This suggests that multi-species analyses of experimentally determined combinatorial TF binding will help identify genomic regions critical for tissue-specific gene control.

## Introduction

The combinatorial binding of transcription factors to DNA define the gene regulatory regions that are essential for achieving spatial and temporal gene expression (*Zinzen et al., 2009*; *Gerstein et al., 2012*; *Hardison and Taylor, 2012*). The rapid increase in empirically determined TF bound motifs (*Badis*

**eLife digest** Stretches of DNA called cis-regulatory modules (or CRMs for short) could help researchers to identify the regions of DNA that are most important for controlling genes. CRMs are regions where multiple transcription factors—proteins that control when and how genes are expressed—bind to DNA. As important biological pathways are often regulated by more than one transcription factor, CRMs are therefore a good target when looking for DNA regions that, if mutated, are likely to cause disease.

If a stretch of DNA performs an important role, it is often conserved throughout evolution. This is often observed for genes that make proteins. Indeed, DNA regions that specify critical amino acids that make up proteins are often conserved across distantly related species. However, unlike the changes made to the amino acid encoding parts of genes, it is currently a challenge to predict which changes in the rest of the genome will affect gene expression.

One reason for this challenge is that transcription factor binding sites are rapidly evolving. This rapid evolution means that strictly comparing DNA sequences between species may fail to identify where transcription factors like to bind in the genome. Numerous experimental efforts have therefore been made to map these sites. These have revealed that there are a huge number of regions in the human genome that can bind transcription factors: hundreds of thousands of sites, far more than there are genes. For this reason, there is a great interest in revealing which of these regulatory regions are critical for maintaining normal levels and timings of gene expression.

Ballester et al. compared the binding sites of four transcription factors responsible for regulating liver function in humans, macaques, mice, rats, and dogs. About two-thirds of these binding sites were found in CRMs. Less than half of the CRMs in humans were also CRMs in another species—but Ballester et al. found that these shared CRMs are predominantly in charge of regulating the essential biological pathways that allow the liver to function correctly. In addition, Ballester et al. identified several examples of disease-causing DNA mutations in shared CRMs that affected the expression of genes that make up pathways such as the blood clotting cascade. Genome-wide association studies also uncovered common variants for liver-related traits that were enriched for the CRMs found in more than one species, further supporting their importance.

As transcription factors work in different ways in different tissues, further studies are now required to expand these observations to organs other than the liver. Future work is also needed to investigate the function of thousands of conserved CRMs whose role in liver gene regulation remains unknown.

et al., 2009; Jolma et al., 2013; Weirauch et al., 2013), sequenced genomes (Goode et al., 2010; Lindblad-Toh et al., 2011; 1000 Genomes Project Consortium et al., 2012), and genome-wide profiling of DNA–protein interactions has given us unprecedented insight into the location of gene regulatory regions in multiple tissue and cell types.

In particular, experimental results obtained by chromatin immunoprecipitation (ChIP), FAIRE, and DNase I footprinting assays in combination with high-throughput sequencing have unmasked what was previously a hidden landscape of active DNA regions (Rhee and Pugh, 2011; Furey, 2012; Neph et al., 2012). The compendium of ChIP-seq determined DNA-binding for 119 different proteins in 72 cell experiments produced by the Encyclopedia of DNA Elements (ENCODE) consortium alone has revealed that the number of TF binding events greatly exceeds the number of genes in the genome and that over 8% of the genome can be bound by at least one TF (ENCODE Project Consortium, 2012). The large number of TF bound genomic regions highlights the growing need for rational strategies for distilling these protein–DNA interactions into functional and non-functional categories. It has recently been shown that unlike TF binding events with high TF occupancy levels (measured by ChIP signal), genomic regions with low TF occupancy levels are not responsible for patterned reporter gene expression in Drosophila (Fisher et al., 2012). It remains to be seen how TF occupancy levels relate to functional gene expression in other species.

Comparing DNA between species has long been employed to identify transcription factor (TF) binding sites that comprise gene regulatory regions (e.g., Tagle et al., 1988; Lindblad-Toh et al., 2011). Indeed, functional reporter gene expression assays have shown that many highly conserved

mammalian non-coding regions serve as developmental limb and nervous system enhancers (*Pennacchio et al., 2006*). In contrast, other tissues including the heart (*Blow et al., 2010*; *May et al., 2012*), liver (*Kim et al., 2011*), and adult brain (*Visel et al., 2013*) possess many functional enhancers that do not show such deep phylogenetic preservation at the DNA level. An increasingly used way to identify tissue and species-specific gene regulatory regions is to compare experimentally determined TF–DNA interactions or histone modifications between species (*Kunarso et al., 2010*; *Mikkelsen et al., 2010*; *Schmidt et al., 2010, 2012*; *Xiao et al., 2012*; *Cotney et al., 2013*; *Paris et al., 2013*). For example, we previously established that the target genes of CEBPA and HNF4A, as identified from gene expression studies of conditional liver TF knockout mice, were enriched for TF binding shared in multiple species (*Schmidt et al., 2010*). Similarly, functional *Drosophila* enhancers are more likely to be found in regions with conserved TF binding events detected by ChIP (*Paris et al., 2013*).

Associating common genetic variation with complex traits is another powerful way to identify functional regulatory DNA in the human genome. Over 80% of the most significant single nucleotide polymorphisms (SNPs) associated with human phenotypes and disease occur within non-coding regions of the genome (*Hindorff et al., 2009*). Recent integrative analyses have shown that open chromatin regions obtained for a specific cell type (e.g., DNase I hypersensitivity sites in T-cells) are enriched for reported GWAS SNPs. Importantly, this GWAS enrichment appeared most significant when the DNAse data was ascertained in a cell type relevant to the phenotype studied (*Maurano et al., 2012*; *Reddy et al., 2012*; *Schaub et al., 2012*). Examples of regulatory DNA mutations that explain differences in disease gene function are increasingly being discovered (e.g., *Musunuru et al., 2010*) and there is tremendous interest in methods that can predict which non-coding variants are of functional consequence (*Schaub et al., 2012*; *Ward and Kellis, 2012a, 2012b*).

To test whether evolutionary conservation of cis-regulatory modules (CRMs) gives insight into human gene regulatory function, we determined transcription factor (TF) binding locations of four liver-enriched TFs in liver tissue from: two primates (human and macaque) estimated to have diverged 29 million years ago; two rodents (mouse and rat) estimated to have diverged 25 million years ago; and dog which diverged during the mammalian radiation along with primate and rodent lineages (*Hedges et al., 2006*).

The liver is a suitable tissue for studying vertebrate gene regulation. It is a relatively homogenous tissue with approximately 75% of the nuclei in the liver coming from hepatocytes (*Marcos et al., 2006*). Both the relative homogeneity and the large cell numbers that can be isolated from diverse organisms under physiologically optimal conditions lend itself well to comparative studies. We focus on four TFs required for liver cell specification and gene function (HNF4A, CEBPA, ONECUT1, and FOXA1) (*Kyrmizi et al., 2006*). Together, several studies have demonstrated that these four TFs work together directly and indirectly to drive liver-specific function (*Plumb-Rudewiez et al., 2004*). Using liver as a model tissue, we demonstrate how a combinatorial analysis of TF occupancy across multiple species can highlight conserved and species-specific biological processes, as well as potential mechanistic actions of disease variants.

## Results

### Determining combinatorial binding in multiple mammalian species

The genome-wide occupancy of four transcription factors (HNF4A, CEBPA, ONECUT1, and FOXA1) was determined in primary liver in five species (*Homo sapiens* [Hsap], *Macaca mulatta* [Mmul], *Canis familiaris* [Cfam], *Mus musculus* [Mmus], and *Rattus norvegicus* [Rnor]) using chromatin immunoprecipitation followed by high-throughput sequencing (ChIP-seq) (*Figure 1*, *Figure 1—figure supplement 1A*, *Figure 1—source data 1*). The antibodies used for the four TFs have been raised against conserved epitopes and have previously been validated in ChIP experiments in mouse and human ChIP studies (*Figure 1—source data 1D*). As expected from previous multi-species ChIP study of CEBPA and HNF4A (*Schmidt et al., 2010*), the known binding motifs for the four TFs was virtually identical between species and occurred close to the ChIP-seq binding summit (*Figure 1—figure supplement 1B,C*).

Similar to what was observed for previous CEBPA and HNF4A ChIP-seq experiments, only a minority of ONECUT1 and FOXA1 bound regions overlapped orthologous, TF-bound regions in a second species, a relationship we refer to here as "shared" TF binding (see *Figure 1A,B*, *Figure 1—figure supplement 2A*). The rapid evolution of TF binding is further supported by comparisons within

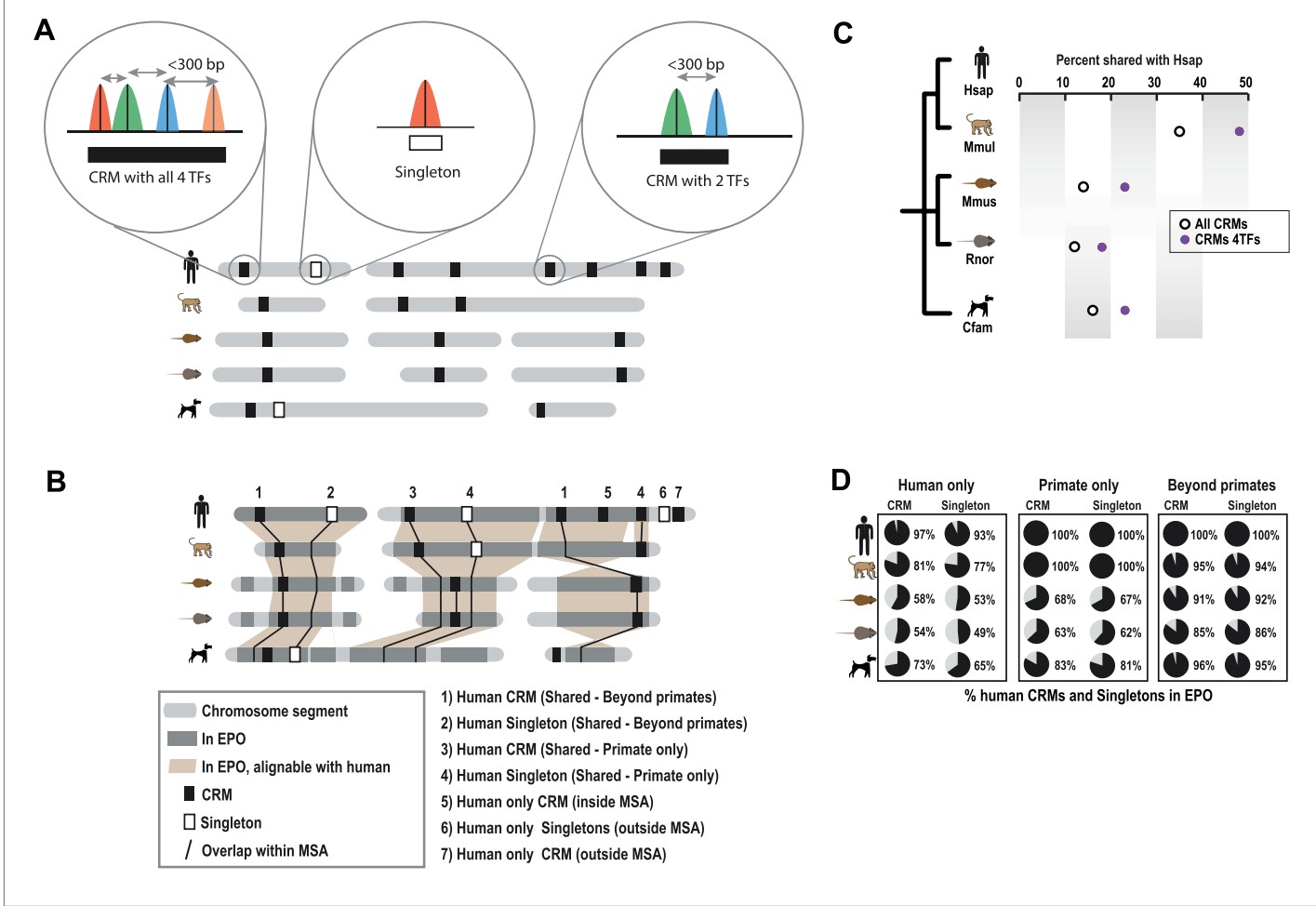

**Figure 1**. Overview of ChIP-seq, CRM construction, and multiple-species comparisons. ChIP-seq peaks were determined for four liver TFs in five mammals. (**A**) CRMs were constructed by merging ChIP-seq peaks whose summits occurred within 300 bp and consisted of at least two distinct TFs. Remaining peaks were designated as singletons. (**B**) Whole genome 9-way EPO multiple sequence alignments (MSA) were used to project CRMs/Singletons across the five species. A CRM was considered shared if its position in the EPO MSA overlapped a CRM in a second species by a minimum of 10 bp. Neither the content nor order of TFs within the CRM was required to be classified as a 'Shared' CRM. A singleton in one species was considered 'Shared' if it overlapped the same TF in a second species. (**C**) Relative to human, the average % of shared CRMs is shown. Human CRMs (comprised of any two TFs) that overlap a CRM from a second species are shown with empty circles. Human CRMs containing at least one of each TF (all 4 TFs) were compared to all identified CRMs in a second species (purple circles). (**D**) The percentage of human CRMs and singletons in different phylogenetic categories that can be found aligned within the EPO MSAs for each of the five species is shown.

The following source data and figure supplements are available for figure 1:

**Source data 1**. Quality control for ChIP-seq, CRM construction, and multi-species comparisons.

**Figure supplement 1**. Summary of ChIP-seq peak number and TF motif enrichment.

**Figure supplement 2**. Pairwise analysis of individual TFs using EPO multiple sequence alignment.

**Figure supplement 3**. A majority of the four liver-enriched TFs cluster into CRMs.

primate and rodent orders that are separated by less than 25 million years (*Springer et al., 2003*). For example, on average, between 21 and 37% of TF binding events in human are found in the orthologous location in macaque and 21–31% between mouse and rat for each of the TFs assayed (*Figure 1—figure supplement 2A*).

Tissue-specific TFs are known to bind in close proximity to form cis regulatory modules (CRMs). Similar to what has been done for multi-TF binding analyses in *Drosophila* (*Zinzen et al., 2009*) and mouse (*Stefflova et al., 2013*), we defined CRMs by clustering at least two proximal heterotypic TF binding events (*Figure 1A*). The number of liver TFs forming clusters falls off sharply when the distance between them is greater than 150 bp, which is less than the average width of the TF bound regions we detected by ChIP-seq (*Figure 1—figure supplement 3*). We built CRMs by merging TF binding events whose summits were within 300 bp of each other (*Figure 1*). Using this summit-based clustering, we found that approximately two thirds of the human liver TF binding events were incorporated into CRMs (*Figure 1—figure supplement 1A*). We found that the shared CRM categories were robust to using a more permissive peak caller or calling peaks on individual biological replicates (*Figure 1— source data 1E*).

As we found for individual TFs, the location of CRMs appears to have evolved rapidly (*Figure 1C*). For example, we found that only ~35% of human CRMs had a CRM in the orthologous macaque genomic region. Similarly, ~32% of mouse CRMs were found as CRMs in the orthologous location in the rat genome (*Figure 1—figure supplement 2C*). This divergence of CRM occupancy was consistent between different lineages separated by the similar evolutionary distances (*Figure 1— source data 1F*), robust to the multiple sequence alignments (MSA) used to detect orthologous CRMs, and also robust to different overlap methods chosen to infer CRM conservation between species (*Figure 1—source data 1G*). *Figure 1D* shows that most (>93%) of human CRMs and singletons we detect are found in the EPO MSA (*Paten et al., 2008*) with macaque, which suggests that the rapid turnover observed between human and macaque CRMs is not due to characteristics of the multiple alignment.

CRMs containing all four TFs are on average more highly shared with a CRM from a second species (e.g., 53% of human CRMs with all four TFs are shared with a macaque CRM), indicating increased selection pressure on higher order combinatorial TF binding (*Figure 1C*, *Figure 1—figure supplement 2C*). TF binding events shared in multiple species are more likely to be found within CRMs (72% of shared human TF binding events are in CRMs vs 27% that are classified as singletons; hypergeometric test, p = 8.48 × 10$^{-238}$). For example, 32 of the 35 CEBPA binding events previously found to be bound in orthologous regions in five vertebrate species (*Schmidt et al., 2010*) fell within CRMs identified in this study.

## Comparative genomic analysis of combinatorial TF binding creates biologically meaningful categories from in vivo ChIP-seq data

To test how combinatorial binding and TF binding conservation relate to liver gene function, we classified our set of human CRMs (n = 31,765) and singletons (n = 43,824) into phylogenetic categories (*Figure 2*, *Figure 2—source data 1*). CRMs were categorized as one of the following: shared only in human and macaque (Primates only, n = 4672); shared in human plus at least one non-primate (Beyond primates, n = 7631); and shared in at least three species (Deeply shared n = 5046) (*Figure 2*). The 43,824 singletons not residing in CRMs (44%) were categorized in the same manner (*Figure 2*).

TF binding events contained in CRMs were enriched for their respective TF's DNA binding motif, in addition to other liver TF binding motifs, including those profiled in this study (*Supplementary file 1*). Supporting this observation, we find that both Deeply shared and human only CRMs overlap significantly with relevant ENCODE genome-wide experimental data sets, including ChIP peaks for HNF4A, FOXA1, and CEBPB in the liver cancer cell line HepG2 (e.g., p < 10$^{-149}$ and p < 10$^{-212}$ for HNF4 respectively; *Supplementary file 1*). The fold enrichment was higher for Deeply shared CRMs than for human only CRMs (e.g., 20.5 vs 11.8 for HNF4A). We also found significant overlap with TFs not tested in our study. For example, 51% of the Deeply shared CRMs and 20% of the human only CRMs overlap binding peaks for SP1 in HepG2 cells (p < 10$^{-149}$ and p < 10$^{-212}$ respectively; *Supplementary file 1*). Again the enrichment for these additional TFs was higher in the Deeply shared CRM category than the human only category (e.g., 17.6 vs 8.4-fold). SP1 and HNF4A have previously been shown to cooperatively regulate gene expression in HepG2 cells (*Sugawara et al., 2007*). In this manner, Deeply shared CRMs can be used to enrich for additional TFs that might play global combinatorial roles in liver gene regulation when used in conjunction with other data sets from related cell types.

A comparison of shared CRMs to human-specific CRMs reveals an increase in the number of liver-related biological pathways, diseases, and known target genes of liver enriched TFs (*Figure 3*). We used

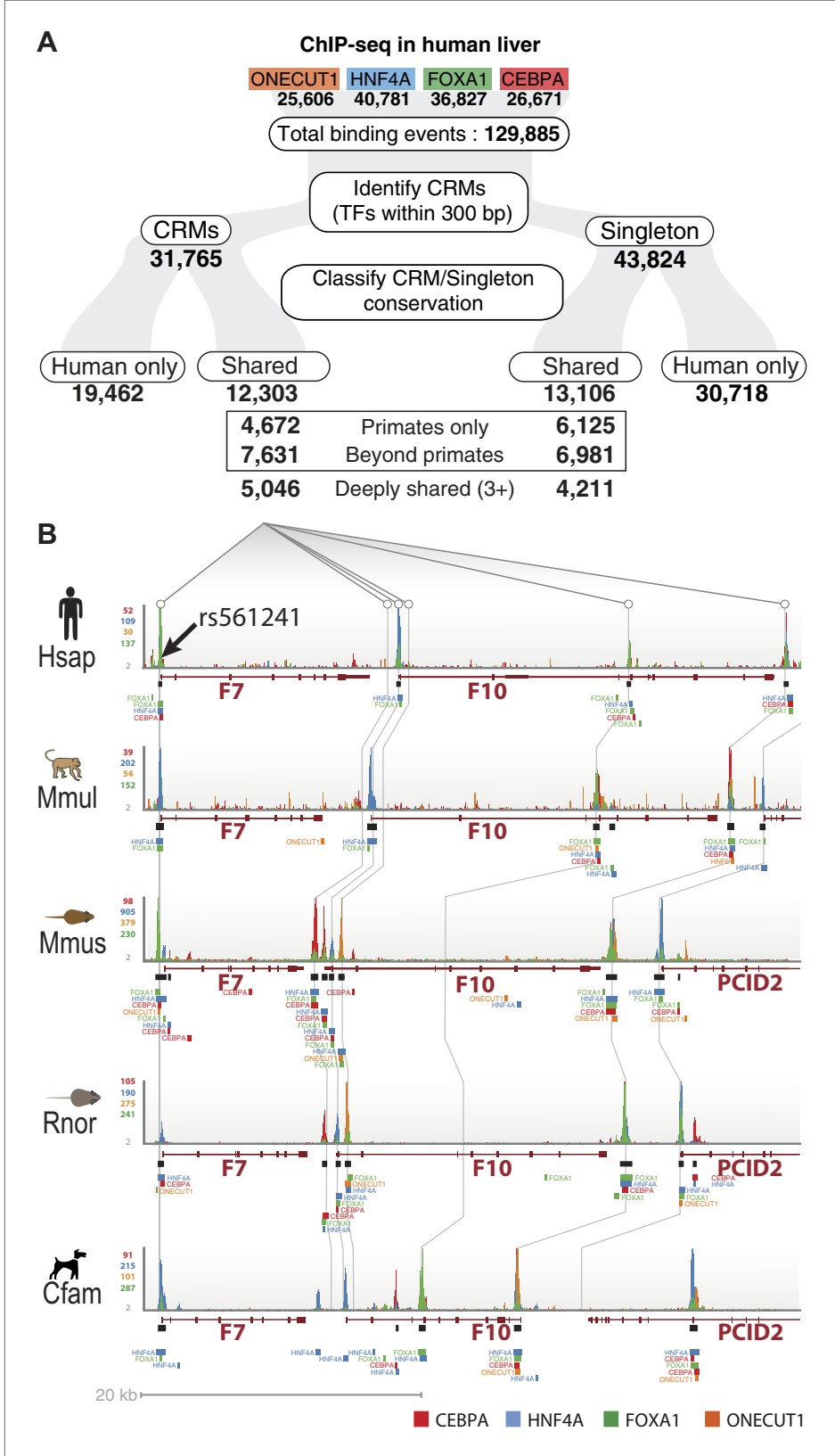

**Figure 2**. Annotation of human regulatory regions using interspecies combinatorial transcription factor binding. (**A**) Human liver ChIP-seq data from ONECUT1, HNF4A, FOXA1, and CEBPA were assembled into CRMs consisting of at least 2 of the 4 TFs. The CRMs or single TFs were then broken down into categories based on their overlap

*Figure 2. Continued on next page*

*Figure 2. Continued*

with ChIP-seq data in macaque, dog, mouse, and rat. Singletons and CRMs were considered shared if they overlapped at least 10 bp with another TF bound region in the EPO multiple sequence alignment (MSA). (**B**) Experimentally determined combinatorial binding at the blood coagulation F7 locus. Raw sequencing reads from ChIP-seq experiments: CEBPA (red), HNF4a (green), ONECUT1 (yellow), and FOXA1 (green) are overlaid and called peaks are displayed for each species. ChIP-seq determined TF binding events were assembled into CRMs (black bars) underneath the enriched regions (peaks). Grey lines are drawn to illustrate shared CRMs using the EPO-MSA.

The following source data is available for figure 2:

**Source data 1**. Table of CRMs and singletons along with the phylogenetic categories they were assigned.

---

the enrichment tool GREAT (*McLean et al., 2010*) to perform functional enrichments. GREAT's default setting assigns TF binding events to a basal region around every gene (5 kb upstream, 1 kb downstream). ChIP-seq peaks that fall within the basal regulatory region of each gene, as well as the genomic sequence that spans between the basal region of that gene and the nearest gene's basal region (within a maximum of 1 Mb) are used to generate functional enrichments. The most significant enrichments that were unique to the shared liver CRMs include: liver disease (Binomial FDR q-value = $4.53 \times 10^{-130}$) from the Disease Ontology database and metabolism of lipids and lipoproteins (q = $2.96 \times 10^{-73}$) from MSigDB Pathway (*Figure 3A,B*, *Figure 3—source data 1A*).

The most significant liver-related enrichments obtained using human only CRMs were for biological oxidations (q = $1.95 \times 10^{-39}$; in MSigDB Pathway). These enrichments were driven by genes involved in the metabolism of xenobiotics by the cytochrome P450 gene family (q = $2.24 \times 10^{-19}$; HGNC gene family database) (*Figure 3A*). Given the liver's major contribution to the detoxification of xenobiotics and the well established species-specificity of the proteins involved in the process (*Gonzalez and Nebert, 1990*), these results suggest that evolutionary filtering of CRMs has the potential to enrich for both conserved and species-specific biological pathways.

Singleton TF binding events were predominantly enriched for their respective motif, but were not enriched for the motifs from the other three TFs profiled in this study (*Supplementary file 1*). Supporting this, comparisons against all ENCODE TF binding data show that for HNF4A, CEBPA, and FOXA1 singletons, the top ChIP-seq peak association in HepG2 cells corresponded to the TF assayed. HNF4A singletons were enriched for FOX family motifs, albeit not the same FOXA1 motif obtained from CRMs and singleton FOXA1 peaks. Comparing normalized sequence read counts in the HNF4A singletons, and HNF4A-containing CRMs lacking FOXA1 peaks, it is clear that pervasive weak FOXA1 ChIP-seq signal occur at HNF4A binding sites (*Figure 4A*). Further supporting this hypothesis is the similarity of FOXA1 to a portion of the HNF4A motif (*Figure 4B*), and a recent study that showed a close association of HNF4A with FOXA1 motifs (*Guo et al., 2012*).

We asked whether CRMs or singletons differ with regards to the quality of their TF binding motifs. Peaks for each TF were scanned using the RSAT tool matrix-scan with the best position weight matrices (PWM) for each TF. We set three p-value threshold cut offs (stringent:$10^{-4}$, moderate:$10^{-3}$ and lenient $10^{-2}$) based on the comparison between the theoretical and empirical PWM weight score distribution observed for each peak collection as previously described (*Medina-Rivera et al., 2011*). As expected, motifs were identified in the vast majority of their corresponding peak set using the lenient motif threshold. Similarly, for both singletons and CRMs, the moderate and stringent motif searches returned the highest fraction motifs in their corresponding peak set. Interestingly, for both the moderate and stringent motif searches, the singleton TF binding sites had a significantly greater fraction of high quality motifs than they did for CRMs (e.g., 56% for singletons vs 33% for CRMs for the stringent cutoff). This trend was observed for all four TFs in this study (*Figure 4—figure supplement 1*, *Figure 4—source data 1*). Our results in primary liver tissue are supported by an integrative analysis of ENCODE cell lines, which showed that active chromatin states are depleted of regulatory motif instances relative to all regions bound by a given TF (*Ernst and Kellis, 2013*).

After collapsing the four TFs into CRMs, there were still over 40,000 singleton TF binding events. We asked if shared singletons show any distinct genomic properties. Shared singleton TF binding events show high DNA constraint (*Figure 3G*), and a larger fraction are found close to the transcription start site of annotated genes compared to the CRMs shared in the equivalent number of species (*Figure 3H*). Unlike the equivalently shared CRMs, shared singleton TF binding events gave

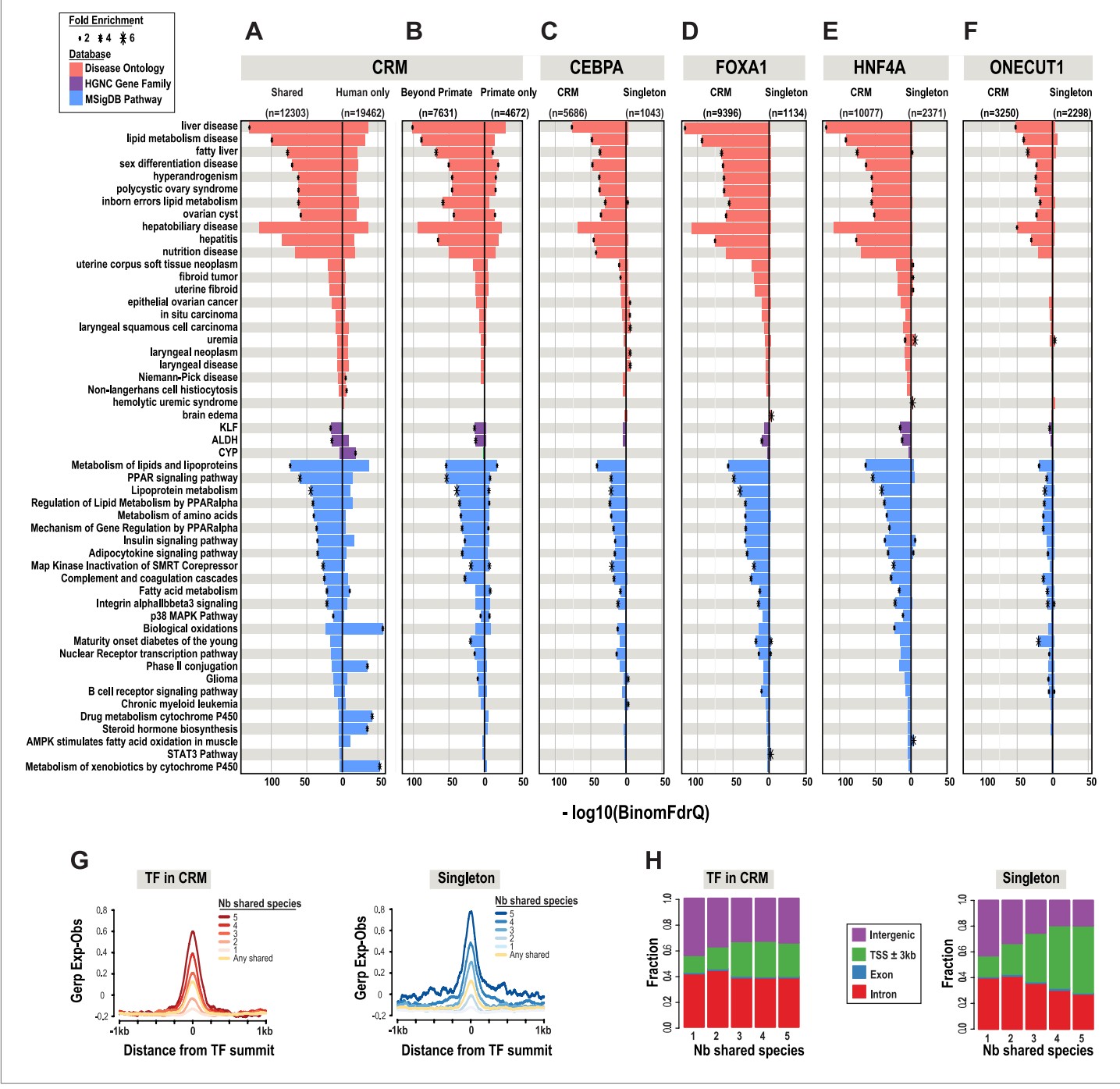

**Figure 3**. Phylogenetic filtering of experimentally determined liver TF binding events yield distinct functional enrichments. Results were obtained using the programming interface for the online enrichment tool GREAT version 2.02 (**McLean et al., 2010**) and plotted with custom R scripts. Up to five of the most significant enrichments obtained for each of the six analyses are listed on the left. The −log10 of binomial Q values for Disease ontology, HGNC gene family, and MSigDB are shown along the x-axis. Bars with a black asterisk indicate significant enrichments using GREAT default parameters (binomial and hypergeometric FDR Q-value significance at P ≤ 0.05 with at least twofold region enrichment). The size of the asterisk is proportional to the fold enrichment obtained for the given database. See **Figure 3—source data 1** for complete list of Q-values, fold enrichments, genes giving the enrichments along with results from additional databases. (**A**) Enrichment analysis of any CRM shared in human plus at least one additional species is shown on the left and human only CRMs are shown on the right (**Figure 3—source data 1A**). (**B**) Human CRMs (left panel) shared in human and at least one non-primate (Beyond Primates) is shown vs Human CRMs (right panel) shared in human and macaque but no other species (Primate only) (**Figure 3—source data 1B**). (**C**) Enrichment analysis of shared CEBPA CRMs and singletons (**Figure 3—source data 1C**). (**D**) Enrichment analysis of shared HNF4A

*Figure 3. Continued on next page*

*Figure 3. Continued*

CRMs and singletons (*Figure 3—source data 1D*). (**E**) Enrichment analysis of shared FOXA1 CRMs and singletons (*Figure 3—source data 1E*). (**F**). Enrichment analysis of shared ONECUT1 CRMs and singletons (*Figure 3—source data 1F*). (**G**) Human TFs in CRMs and Singletons were categorized by the number of species in which they are shared with. Profiles of constrained elements (sequence conservation) in a 1-kb window around CRMs or singletons were calculated using GERP scores from the 29-way multiple sequence alignments. (**H**) Genomic location of CRMs and Singletons. Proportion of single TFs located near transcription start sites (TSS) increases to >50%, but remains stable for CRMs at ~20%.

The following source data are available for figure 3:

**Source data 1**. Functional enrichment results obtained for CRMs and singletons using GREAT.

fewer and less significant enrichments that than the equivalently shared CRMs (*Figure 3C–F*). Relative to CRMs, the enrichments unique to shared singletons, did not appear to be overtly liver specific. For example, no significant liver disease ontology enrichments were found for shared singleton CEBPA binding events; however, several cancer disease enrichments from various tissues, such as in situ carcinoma ($q = 1.91 \times 10^{-5}$), were obtained (*Figure 3C*).

One consideration about comparing singletons to CRMs is that singleton TF binding events are likely to become CRMs as more factors are tested and more peaks are called (see *Figure 1—source data 1E* for a comparison of the stability singleton and CRM categories). Nonetheless, by focusing on the singletons that remain singletons in orthologous regions in two or more species, we have been able to detect distinct genomic properties that warrant future study.

## Shared combinatorial TF binding associates with highly expressed liver-specific genes

To compare the functional properties of shared and species-specific CRMs/singletons, we then looked at how combinatorial binding and evolutionary constraint correlated with gene expression (*Figure 5*). Human TF binding events in CRMs and singletons were categorized by the number of species they were shared in and then associated with the nearest gene. Human liver mRNA expression level of the nearest gene was determined by RNA-seq (*Kutter et al., 2011*). Genes nearest to human-specific singletons and CRMs were not significantly different in their expression levels ($p = 0.221$). In contrast, gene expression levels near TFs in shared CRMs were significantly higher than those near shared singleton-associated genes ($p = 2.4 \times 10^{-16}$) (*Figure 5A*). This striking p-value is due to several CRMs being found close to highly expressed liver genes including albumin, fibrinogen (FGA, FGB, FGG), and several acute phase response genes (e.g., CRP, SAA1 etc). We therefore broke down each CRM and singleton by transcription factor, which still revealed a significant difference between genes close to Deeply shared CRMs relative to singletons ($p < 1 \times 10^{-3}$; *Figure 5—figure supplement 1*). Using a reference transcription data set that comprises RNA-seq data for liver and 15 additional human tissue types (E-MTAB-513), we confirmed the above observation and found that the gene expression association with liver CRMs, and to a lesser extent singletons, was tissue-specific (*Figure 5B*). In sum, Deeply shared CRMs are associated with genes that are highly expressed in a distinctly liver-specific manner.

## Shared CRMs are enriched for tissue-specific biological function

The number of reproducibly bound human regions obtained by ChIP-seq often exceeds the number of genes in the genome and so may require prioritization before experimental validation. Ranking ChIP-seq peak regions based on peak enrichment scores is one logical way to prioritize ChIP-seq peaks (*Fisher et al., 2012*). We compared the pathway enrichments for all shared CRMs containing a specific TF (e.g., the 6278 HNF4A-containing CRMs shared between human and at least one non-primate) vs the equivalent number of CRMs ranked by the best ChIP-seq peak enrichment score of that specific TF (e.g., the top 6278 HNF4A CRMs ranked by HNF4A peak score).

As expected, the HNF4A CRMs ranked by peak intensity showed higher read counts than the shared set of CRMs and both CRM sets showed strong, centralized motif enrichments (*Figure 6A*). However, by using overlap of CRMs in the EPO multiple sequence alignment as a filter, we found that shared CRMs give an increased number of significant enrichments using the ChIP-seq enrichment analysis tool, GREAT (*Figure 3—source data 1*). As observed for our collection of shared CRMs, the most significant enrichments for the shared CRMs are related to liver metabolic processes and disease (*Figure 6B*). Similar results were obtained by performing this comparison from the perspective of the other three TFs (*Figure 6B*).

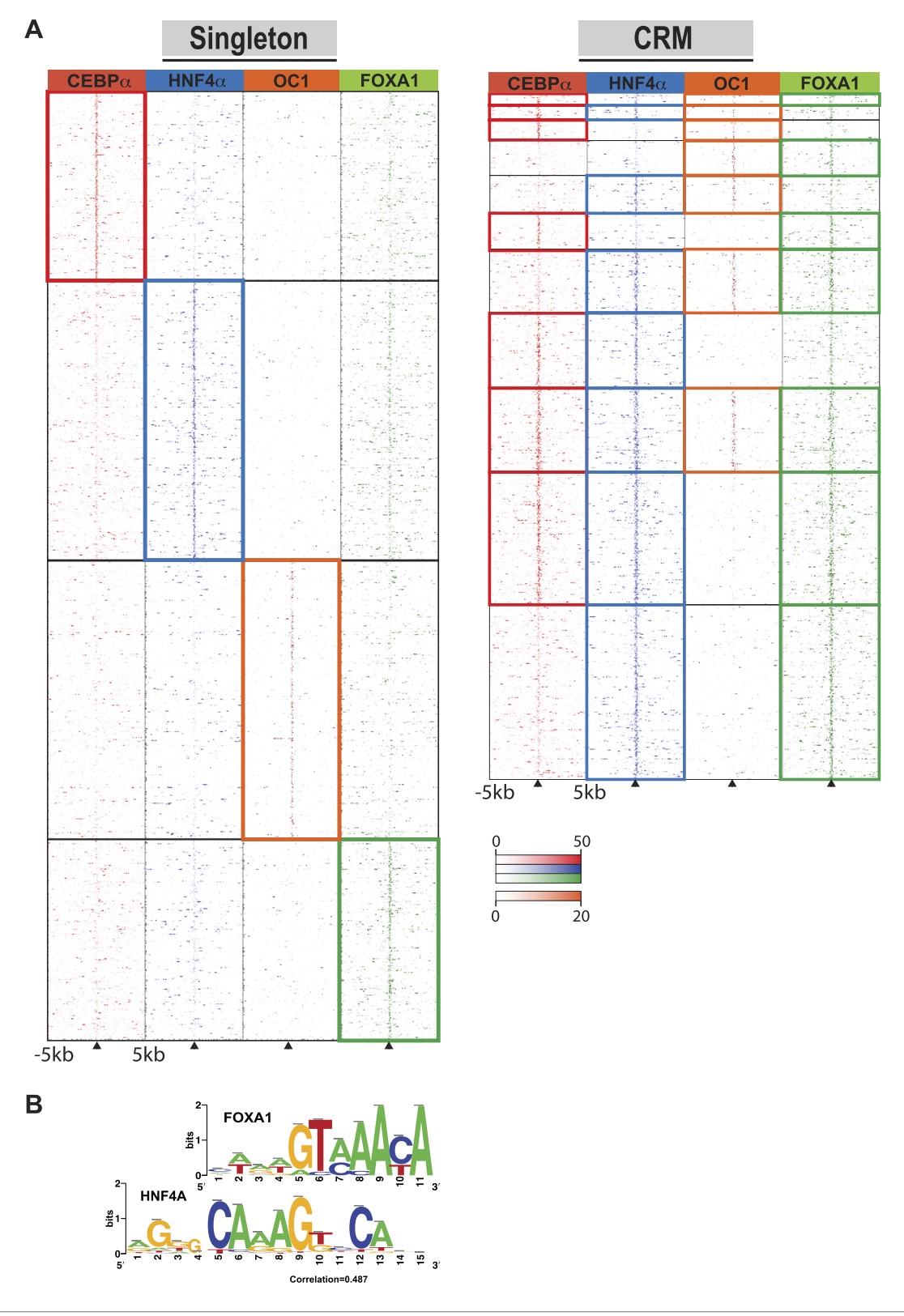

**Figure 4**. Comparison of TF occupied regions classified as CRMs and singletons. (**A**) Regions of ±5 kb are represented around the center of CRMs or singletons. Reads centered on the summit of each TF are counts subtracted by input reads in 100 bp bins plus and minus 5 kb from the summit. Colored boxes indicate CRMs or singletons where a peak was called for a given factor: CEBPA (red), HNF4A (blue), ONECUT1 (orange), and FOXA1 (green).
*Figure 4. Continued on next page*

*Figure 4. Continued*

Looking at read counts for all four factors reveal that many of the HNF4A singleton in fact have weak FOXA1 signal. (**B**) Alignment of FOXA1 de novo ChIP-seq motif to the HNF4A motif. Motif comparison (alignment) was performed using *compare-matrices* from RSAT. The program calculates the correlation between two matrices shifting positions; the correlation is normalized based on the width of the alignment to avoid high correlation based on few flanking positions.

The following source data and figure supplement are available for figure 4:

**Source data 1**. Comparison of motif matches between CRMs and singletons. Chi-square test for differences between the number of peaks associated with CRMs and singletons, for each TF, that contained at least one predicted motif using three different p-value thresholds for motif scanning: stringent ($10^{-4}$), moderate ($10^{-3}$) and lenient ($10^{-2}$). Blue shadows highlight siginficnat p-values.

**Figure supplement 1**. Comparison of stringent motif matches between CRMs and singletons.

## Lead SNPs from liver-related GWAS overlap Deeply shared CRMs

In order to determine whether shared CRMs associate with human diseases and phenotypes, we asked: (1) if the phenotype-associated single nucleotide polymorphisms (SNPs) in a curated collection of GWAS (described here as 'lead SNPs'; *Hindorff et al., 2009*) overlap our CRMs/singletons in a liver-related manner; and (2) whether curated collections of these SNPs were enriched for shared CRMs (*Figure 7*, *Figure 7—figure supplement 1*).

Using hypergeometric testing, we found that liver-related GWAS lead SNPs were enriched for nearby (±2.5 kb from a SNP) Deeply shared CRMs. For example, we found lead GWAS SNPs related to liver (p = $8.38 \times 10^{-6}$; 3.7-fold enriched), blood lipid (p = $9.68 \times 10^{-5}$; 3.3-fold enriched), and drug response (p = $1.43 \times 10^{-4}$; 5.1-fold enriched) categories were all enriched for shared CRMs (*Figure 7*, *Figure 7—source data 1A*). Repeating this analysis using linkage disequilibrium (LD) measures ($r^2 \geq 0.8$) to define the boundaries of each GWAS SNP gave similar results. For example, liver-related GWAS SNPs were enriched for Deeply shared CRMs when LD was taken into consideration (p = $3.20 \times 10^{-9}$; 2.4-fold enriched; *Figure 7—figure supplement 1*, *Figure 7—source data 1B*). These enrichments were not found when using two distinct null models (*Figure 7—source data 1C,D*).

We then asked if any specific disease traits, as written in the NHGRI GWAS catalog, were enriched for shared CRMs. We found that for the 2.5 kb window analysis, only LDL cholesterol significantly enriched for Deeply shared CRMs (p = 0.037; 4.54-fold enriched; *Figure 7—source data 1E*), whereas the LD window analysis revealed 11 disease traits that were enriched for Deeply shared CRMs (see *Table 1*; *Figure 7—source data 1F*). For example, enrichments driven by lead SNPs for LDL cholesterol (blood lipid category, p = $4.84 \times 10^{-5}$; 3.7-fold enriched) involved several loci including *TRIB1, ABCG8, APOB, SORT1, TOMM40, APOA5* and *HNF1A*. Lead SNPs for fibrinogen (liver category, p = $3.40 \times 10^{-3}$; 3.4-fold enriched) occurred in LD with the fibrinogen locus. C-reactive protein (liver category, p = $1.17 \times 10^{-4}$; 4.0-fold enriched) enriched for Deeply shared CRMs near CRP itself in addition to *RORA, MLXIPL, HNF1A, BAZ1B*, and *IRF1* loci (*Table 1*; *Figure 7—source data 1G*).

In order to explore the functional relevance of these findings, we looked for annotated regulatory SNPs in the RegulomeDB database (*Boyle et al., 2012*) within the 1020 CRM or single TF regions we found to be within 2.5 kb of a lead GWAS SNP. Of these 1020 regions, 753 contained at least one variant, 90% of which showed evidence of regulatory potential in RegulomeDB (*Supplementary file 2*). In particular, 317 of these 753 regions had TF binding in orthologous regions in additional species, making them rational candidates for future functional exploration.

We also asked whether the collection of recently identified 'super-enhancers' (*Hnisz et al., 2013*), which were enriched for disease loci in a tissue-specific manner, would also enrich for liver-related lead GWAS SNPs. Our analysis supported the association between super-enhancers and immune system related GWAS SNPs reported by Hnisz et al. (with the highest enrichment for immune cell GWAS lead SNPs found in 'super-enhancers' in cell line CD20, p = $3.24 \times 10^{-6}$). However, unlike what we found for shared CRMs in liver, super-enhancers obtained from the liver cancer cell line HepG2 were not enriched by the liver-related lead GWAS SNPs (*Figure 7—figure supplement 2*). This observation may be related to biological differences between primary liver tissue and HepG2.

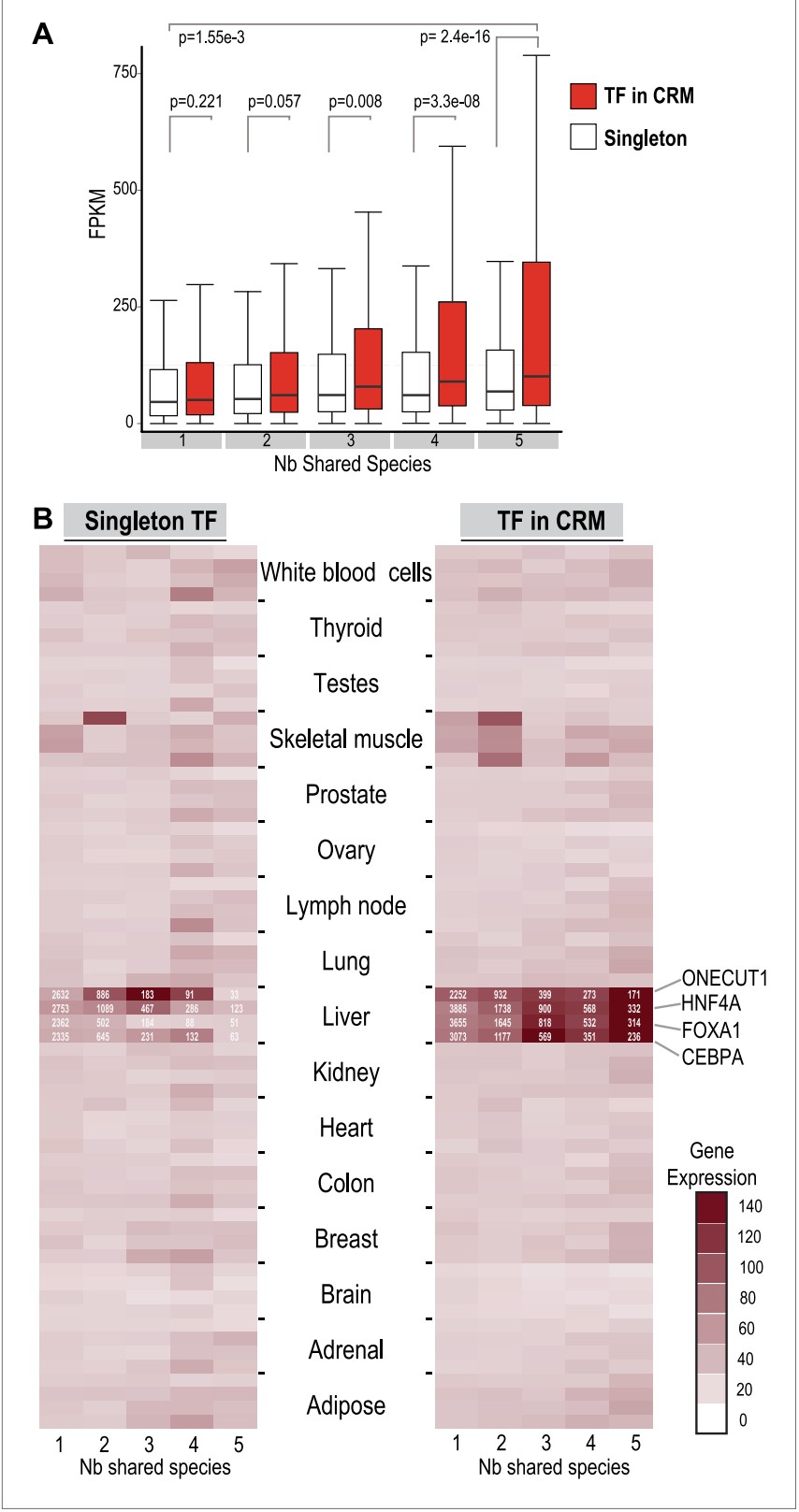

**Figure 5**. TFs in Deeply shared CRMs are near genes highly expressed in a tissue-specific manner. (**A**) Association of shared TFs in CRMs and Singletons with human gene expression obtained by RNA-seq in human liver (**Kutter et al., 2011**; E-MTAB-424). TFs in CRMs or Singletons were assigned to the nearest gene, and the FPKM

*Figure 5. Continued on next page*

*Figure 5. Continued*

(Fragments Per Kilobase of exon per Million reads) was recorded. In contrast to Singletons, TFs in Deeply CRMs are associated with highly expressed genes (adjusted p-values shown). The numbers of target gene associations for the singletons and CRMs in categories 1 to 5 are: 19354(S), 32706(CRM); 6325(S), 14669(CRM); 1935(S), 5755(CRM); 1005(S), 3292(CRM); and 459(S), 2530(CRM). (**B**) Comparison with CRMs and Singletons to a reference mRNA-seq data from 16 human tissues (E-MTAB-513) further shows that relative to singletons, liver-specific CRMs are highly expressed in liver, and that each TF contributes to this specificity. The number of gene associations for each category in the liver data is shown in white text within the heat map.

The following figure supplement is available for figure 5:

**Figure supplement 1**. Association of shared TFs in CRMs and Singletons with human gene expression obtained by RNA-seq in human liver (*Kutter et al., 2011*; E-MTAB-424) broken down by the transcription factor in the CRM: (**A**) HNFA; (**B**) CEBPA; (**C**) FOXA1 and (**D**) ONECUT1.

## Shared CRMs are disrupted in known liver-related diseases affecting blood coagulation and lipid regulation

To further investigate the functional role of shared CRMs and to test the hypothesis that disruption of conserved combinatorial binding can lead to human disease, we overlapped our CRMs and singletons data with manually curated regulatory mutations directly linked to human disease (Human Genome Mutation Database Professional version; HGMD) (*Stenson et al., 2012*). This database contains the most comprehensive set of curated and functionally validated mutations that have occurred in human regulatory DNA regions and have led to a change in disease gene expression. A total of 157 genes associated with regulatory mutations overlapped our human ChIP-seq data, 106 of which were in our CRMs (*Table 2*; *Supplementary file 3*). The Deeply shared CRMs overlapped a set of 47 genes associated with regulatory mutations. These 47 genes were clearly enriched for two liver-related biological pathways: coagulation and complement factors (p = $9.34 \times 10^{-6}$; 24-fold enrichment) and lipid homeostasis (p = $6.93 \times 10^{-5}$, 36-fold enrichment) (*Figure 8—source data 1A,B*). We found multiple disease-causing regulatory mutations overlapping our CRMs at promoters of seven critical genes in the coagulation pathway (*FGA*, *FGB*, F7, F9, F10, F11, F12; *Figure 8*, *Figure 8—source data 1*). While many of these mutations have been individually known for decades, this is the first time they have been put in context of a regulatory network consisting of these liver-enriched TFs. Furthermore, repeated observation of rare mutations in the promoter regions of several of the blood coagulation proteins correspond to critical positions in predicted DNA binding motifs for the liver-enriched TFs. For example,

1. Mutations leading to Factor VII deficiency and severe bleeding disorders occur in a promoter region harboring a CRM shared in all five species (*Figure 8—figure supplement 1A*). Several mutations (including SNP variant rs561241) within a conserved HNF4A motif have been shown to perturb F7 transcription leading to hemophilia (*Zheng et al., 2011*).
2. A critical highly conserved CRM localizes to the F9 promoter (*Figure 8—figure supplement 1B*). Multiple mutations within this region are associated with defective expression of F9 and clinical hemophilia (*Giannelli et al., 1998*). Several of these mutations have previously been shown to disrupt the binding of CEBPA and HNF4A (*Crossley and Brownlee, 1990*; *Crossley et al., 1992*; *Reijnen et al., 1993*; *Giannelli et al., 1998*). We recently demonstrated that ONECUT1 binds to the −6 site of F9 in human and mouse (*Funnell et al., 2013*). Inspection of the multiple species alignment suggests the same arrangement and spacing of CEBPA, HNF4A, and ONECUT1 motifs in the human and macaque CRMs, whereas the mouse and rat ONECUT1 motifs are predicted to begin three base pairs upstream.

In this analysis of known functional and disease causing regulatory mutations from the HGMD database, it is worth noting that most of the examples we found overlapping Deeply shared liver CRMs resided close to the TSS. As most TF binding of the factors we profiled occur outside of proximal promoters, it is likely that many more human mutations that regulate genes through long-range interactions remain to be found. Just as gene sequencing has uncovered a diverse array of mutations for most disease genes, we expect that disruption of conserved TF bound regions will be found to have pathological consequences.

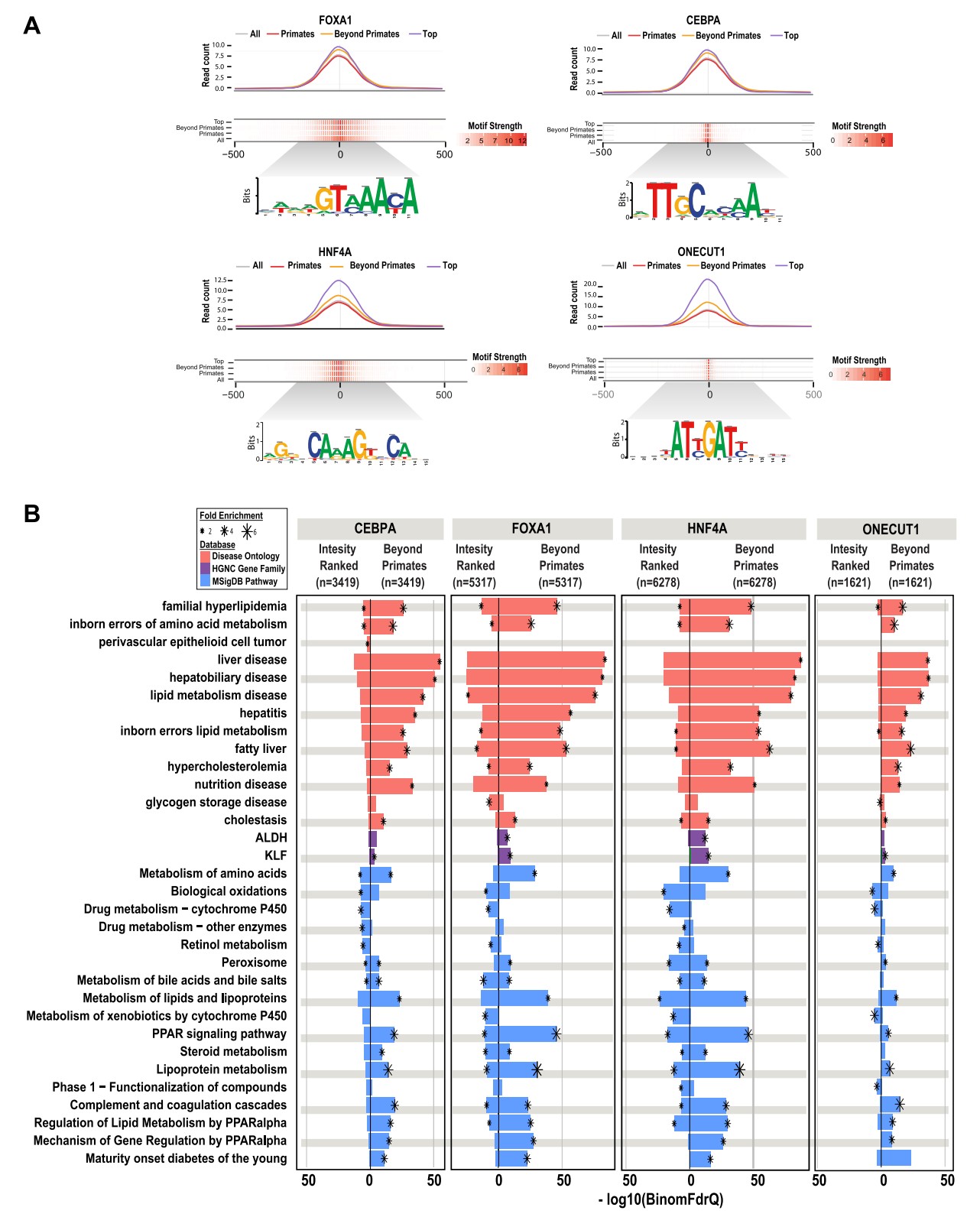

**Figure 6**. Shared HNF4A CRMs unravel more liver related functional classes than do the equivalent number of CRMs with the best peak enrichment scores. CRMs containing each TF were analyzed separately. (**A**) Read count and motif binding weight scores were calculated for: (1) all CRMs (All); (2) CRMs shared in human and at least one additional non-primate (Beyond primates); (2) human CRMs shared in macaque only (Primates); and
*Figure 6. Continued on next page*

*Figure 6. Continued*

(3) the equivalent number of CRMs (equal to the number of Beyond primate CRMs) ranked by the SWEMBL peak intensity score for the TF in question (Top). (**B**) Functional enrichments were performed using GREAT comparing the Beyond primate category to the top ranked category. The top five enrichments for all comparisons performed were collected and the enrichments, if available are plotted. Databases used for GREAT enrichment analyses are indicated by color and are ranked according to the −log10 binomial FDR q-values plotted on the x-axis. Significant enrichments are labelled with an asterisk which is sized according to fold enrichment of the given database category.

## Discussion

There is a great interest in identifying and mechanistically characterizing regulatory SNPs. Such approaches require limiting the genomic search space, finding the function of identified mutations, associating the mutation with target gene(s) and addressing the tissue-specificity inherent in transcriptional regulation. To limit the search space, the genome must be distilled to the active functional regions where mutations are likely to be pathological. To determine the function of mutations in gene control regions, the trans-acting molecules (e.g., transcription factors) whose regulation of a particular

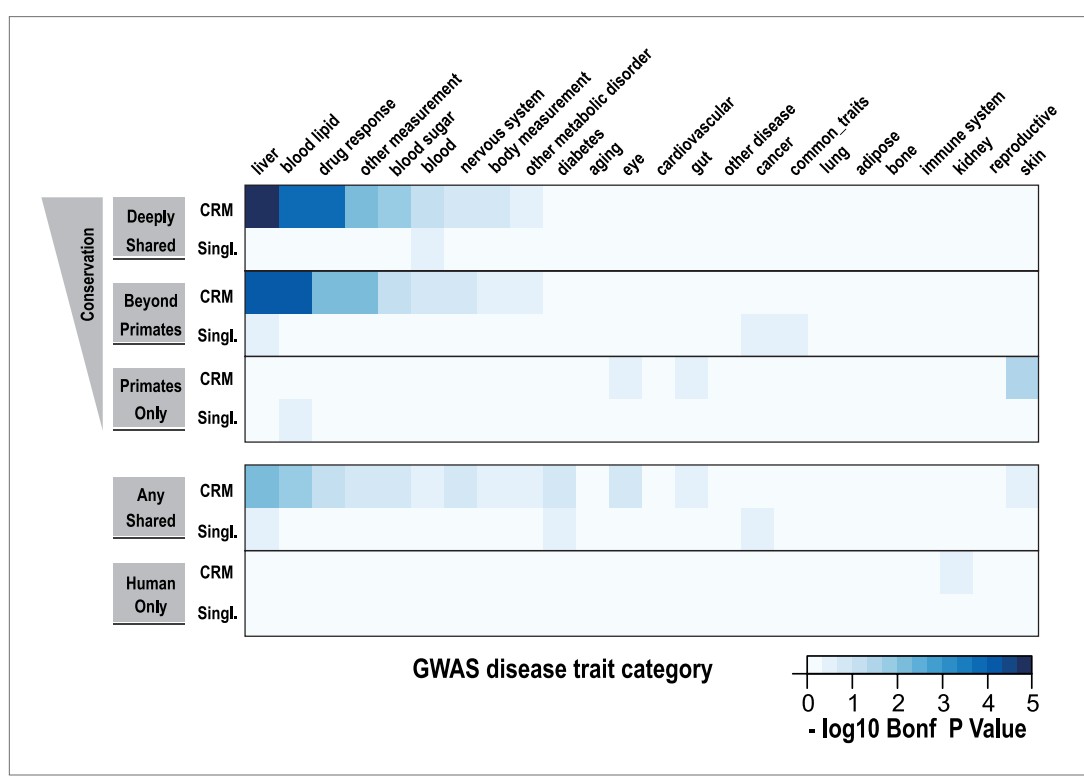

**Figure 7**. Using a window of ±2.5 kb, lead SNPs obtained by GWAS were enriched for shared CRMs in a tissue/disease specific manner. Heatmap representation of the −log10 of Bonferroni corrected p-values from hypergeometric testing for enrichment of CRMs or single TFs (broken down into categories related to their degree of conservation) by lead GWAS SNPs obtained from the NHGRI catalog (*Hindorff et al., 2009*). The NHGRI catalog disease traits were summarized into 25 categories prior to enrichment. Each GWAS lead SNP was given a ±2.5-kb window prior to identifying overlapping CRMs/singletons.

The following source data and figure supplements are available for figure 7:

**Source data 1**. Full tables of 2.5 kb and LD GWAS enrichments performed in *Figure 7* and *Table 1*.

**Figure supplement 1**. Using linkage disequilibrium (r² ≥ 0.8), GWAS lead SNPs were enriched for shared CRMs in a tissue/disease specific manner.

**Figure supplement 2**. Super-enhancer enrichments obtained by GWAS lead SNPs.

**Table 1.** Table of GWAS Disease Traits that significantly enriched for Deeply shared CRMs

| Disease trait (NHGRI) | Category (this study) | Number of Deeply shared CRMs | Deeply shared CRM enrichment (adjusted p-value) | Fold enrichment | Closest genes |
|---|---|---|---|---|---|
| Other metabolic traits | other measurement | 11 | 4.42E-05 | 6.42 | CRP, HNF1A, PANK1 |
| LDL cholesterol | blood lipid | 20 | 4.84E-05 | 3.71 | TRIB1, ABCG8 PSRC1, DOCK7, APOB, HNF1A, LDLR, TOMM40, HNF1A, APOA5 |
| C-reactive protein | liver | 17 | 1.17E-04 | 3.97 | MLXIPL, RORA, CRP, HNF1A, TOMM40, BAZ1B, IRF1 |
| D-dimer levels | liver | 11 | 6.87E-04 | 4.99 | NME7, FGG, EDEM2, FGA |
| Fibrinogen | liver | 15 | 3.40E-03 | 3.40 | FGB, FGA, FGG |
| Lung cancer | cancer | 14 | 4.39E-03 | 3.46 | C2, CRP, HSPA1A, TP63 |
| Mean corpuscular hemoglobin | blood | 8 | 8.57E-03 | 5.02 | GCDH, USP49, RCL1, SLC17A1, TFRC, MPST |
| Protein quantitative trait loci | other measurement | 14 | 2.49E-02 | 2.93 | CRP, IFT81, BCO2 |
| Serum markers of iron status | other measurement | 13 | 2.67E-02 | 3.03 | TCP1, MRPL18 , TF, SLC17A1, HIST1H4C, MPST, GHR |
| Triglycerides | blood lipid | 12 | 2.84E-02 | 3.16 | TRIB1, MLXIPL, DOCK7, BAZ1B, GALNT2, TRIB1, APOA5 |
| Select biomarker traits | other measurement | 8 | 3.89E-02 | 4.08 | CRP, OR10J5 |

Lead GWAS SNPs and their associated Disease Traits were obtained directly from the NHGRI catalog (**Hindorff et al., 2009**). An LD window ($r^2 \geq 0.8$) around each SNP was obtained and regions with identical Disease Traits were collapsed into a single interval. These Disease-Trait–associated intervals were then intersected with all CRMs and Singleton categories as in **Figure 7**. This table shows Disease Traits that were significantly enriched for Deeply shared CRMs. The summarized disease category used in **Figure 7**, the number of Deeply shared associated CRMs, the Bonferroni corrected p-values from the hypergeometric test, fold enrichment of Deeply shared CRMs, and the nearest gene to the Deeply shared CRM, if it is protein coding, are shown. **Figure 7—source data 1G** contains detailed enrichment information including SNP ID and primary GWAS publication (PMID).

locus is disrupted need to be identified. Finally, because transcriptional regulatory networks are highly tissue-specific, regulatory regions can only be accurately done within the tissue concerned.

In this study, we demonstrated that genomic regions harboring shared combinatorial TF binding were enriched in a tissue-specific manner for essential biological pathways, common DNA variants associated with complex traits, and regulatory DNA mutations associated with rare diseases. In addition to the requisite measures of ChIP-seq data quality and reproducibility, it could be argued that combinatorial binding of transcription factors, relevant to the cell type of interest, increases the likelihood that the region identified is biologically important. However, more than half of the human liver TF binding sites detected in our study are in CRMs. In many cases, this makes CRM clustering on its own an insufficient filter for prioritizing TF binding sites for further study. Given the rapid turnover of TF binding within a mammalian lineage (**Stefflova et al., 2013**), requiring combinatorial binding to be shared in multiple mammals is a strict filter that provides evidence that these biochemical interactions are under selection, even if the percent identity in the sequence alignments themselves is not exceptional. Supporting this idea, we found the percentage of CRMs shared between human and macaque (~35%) and mouse and rat (~34%) is considerably lower than the average human-rhesus sequence identity of ~90% (**Rhesus Macaque Genome Sequencing and Analysis Consortium et al., 2007**) or the mouse-rat identity of ~93% (**Gibbs et al., 2004**).

Although insightful, the costs and challenges of performing comparative combinatorial TF binding in multiple species, tissues, developmental stages, and environmental conditions limits its widespread use as a method for finding enhancers. Computational strategies that forgo strict DNA constraint for more flexible criteria, such as shared clusters of motifs, show great promise (**Gordân et al., 2010**). For example, a recent computational strategy that relied upon conservation of clusters of motifs, rather than conserved DNA sequence, was able to fine map regulatory SNPs in select GWAS loci (**Claussnitzer et al., 2014**). The results and the strong functional enrichments, we observe with shared combinatorial binding further support such approaches.

**Table 2.** HGMD disease variants falling within motifs of shared CRMs and singletons

| | Coordinates | Gene name | HGMD (regulatory, disease mutations) | Disease mutations |
|---|---|---|---|---|
| **Shared CRM** | 3:30622781-30623121 | TGFBR2 | Marfan syndrome II | 1 |
| | 3:37009728-37010105 | MLH1 | Colorectal cancer, non-polyposis | 8 |
| | 3:95175129-95175686 | PROS1 | Protein S deficiency | 2 |
| | 3:172227012-172227583 | SLC2A2 | Diabetes | 1 |
| | 4:187423808-187424193 | F11 | Factor XI deficiency | 1 |
| | 5:176769019-176769427 | F12 | Factor XII deficiency | 2 |
| | 7:75769704-75769860 | HSPB1 | Amyotrophic lateral sclerosis | 1 |
| | 9:35647685-35648108 | RMRP | Cartilage-Hair hypoplasia* | 60 |
| | 9:103237817-103238177 | ALDOB | Fructose intolerance | 1 |
| | 11:57121333-57121845 | SERPING1 | Angioneurotic oedema | 3 |
| | 11:116213420-116213833 | APOA1 | Apolipoprotein A1 deficiency; Atherosclerosis with coronary artery disease | 2 |
| | 12:119900361-119900931 | HNF1A | Diabetes | 9 |
| | 13:112807935-112808278 | F7 | Factor VII deficiency | 15 |
| | 17:39777939-39778164 | GRN | Amyotrophic lateral sclerosis; Frontotemporal dementia | 2 |
| | 19:11060834-11061300 | LDLR | Hypercholesterolaemia | 23 |
| | 19:40465042-40465277 | HAMP | Haemochromatosis | 2 |
| | 19:50140788-50141367 | APOC2 | Apolipoprotein C2 deficiency | 1 |
| | 20:42417527-42417906 | HNF4A | Diabetes | 13 |
| | X:138440311-138440689 | F9 | Haemophilia B | 22 |
| **Shared singleton** | 1:55277551-55277787 | PCSK9 | Hypercholesterolaemia, autosomal dominant | 1 |
| | 2:47483486-47483635 | MSH2 | Colorectal cancer, non-polyposis | 1 |
| | 5:147191404-147191657 | SPINK1 | Pancreatitis | 5 |
| | 11:107598900-107599060 | ATM | Ataxia telangiectasia | 1 |
| | 17:3486175-3486485 | CTNS | Cystinosis | 2 |
| | 17:27840752-27841062 | CDK5R1 | Mental retardation | 1 |
| | 19:54160105-54160315 | FTL | Cataract, bilateral | 1 |
| | X:66680386-66680665 | AR | Prostate cancer | 1 |
| **Human only CRM** | 1:113300254-113300535 | SLC16A1 | Exercise-induced hyperinsulinism | 1 |
| | 1:224075148-224075491 | EPHX1 | Hypercholanaemia | 1 |
| | 3:170965492-170965762 | TERC | Aplastic anaemia; Dyskeratosis congenita; Myelodysplastic syndrome | 3 |
| | 8:64161127-64161334 | TTPA | Ataxia, isolated vitamin E deficiency | 1 |
| | 10:27429312-27429620 | ANKRD26 | Thrombocytopaenia | 12 |
| | 13:59636050-59636246 | DIAPH3 | Auditory neuropathy | 1 |
| | X:146800975-146801150 | FMR1 | Fragile X mental retardation syndrome | 3 |
| | X:153643911-153644318 | DKC1 | Dyskeratosis congenita, X-linked | 1 |

HGMD disease variants falling within shared CRMs, shared singletons and human only CRMs. The number of unique regulatory mutations designated as 'disease-mutations' in the HGMD database recorded within each CRM or singleton is shown.
*RMRP is a non-coding RNA that is found in HGMD associated to several related diseases.

It is also becoming clear that integrative approaches to enhancer discovery can outperform predictions made using single criteria (*Erwin et al., 2014*). We compared our CRM and singleton data to enhancer predictions made by a recent integrative approach (EnhancerFinder; *Erwin et al., 2014*). EnhancerFinder is trained on experimentally verified human VISTA enhancers and utilizes evolutionary conservation, DNA motifs, and functional genomics data (such as p300 and histone

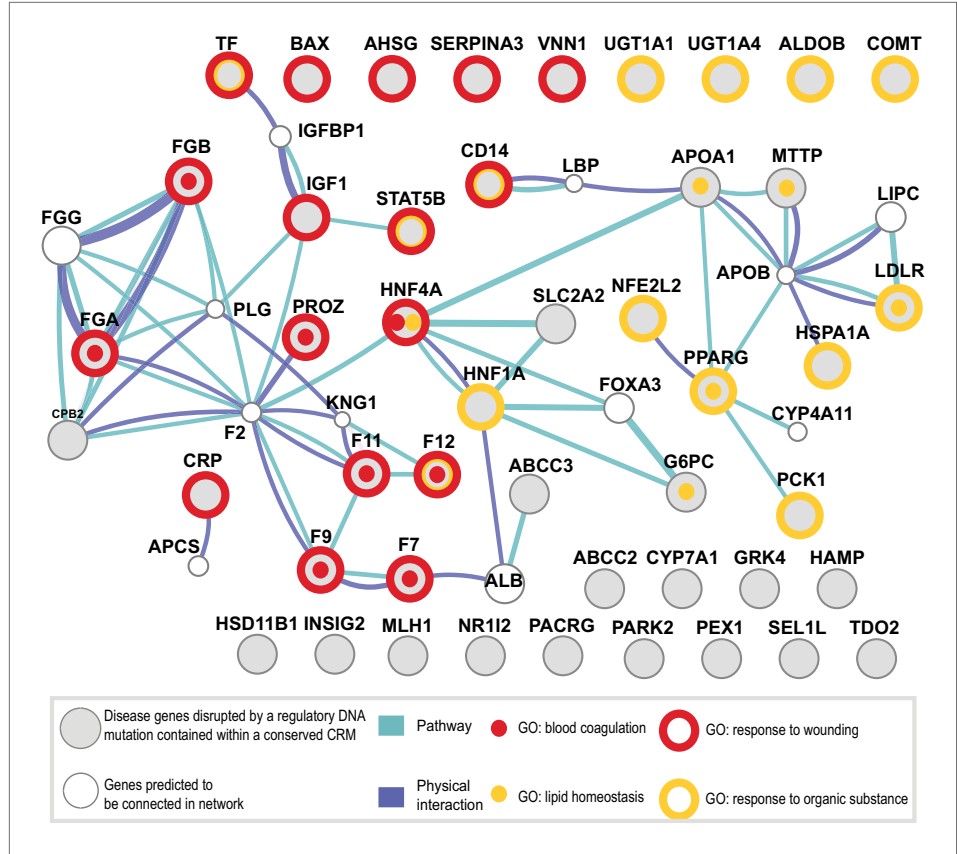

**Figure 8**. Shared CRMs link TF binding with disease-causing variants in coagulation and lipid regulation in the liver. Human CRMs that had TF binding in syntenic regions in at least two additional species (n = 5046) were intersected with the HGMD database. All protein coding genes associated with a regulatory mutation were analysed. Relationships among these genes were investigated and a representative analysis obtained using GeneMANIA ('Materials and methods'). Genes (large grey circles) are connected by pathways and protein–protein interactions are shown. The smaller white circles are genes predicted by GeneMANIA to be in the network. The 47 unique genes were associated into 35 clusters using DAVID (*Huang da et al., 2009*). Eight Gene Ontology terms from the 35 clusters had an adjusted p-value of less than 0.005 (*Figure 8—source data 1A*; *Supplementary file 3*). 4 of the 8 significant GO categories containing the most genes are illustrated: response to wounding (open red circle, $p = 3.16 \times 10^{-9}$; 9.9-fold enriched); blood coagulation (red dot, $p = 9.34 \times 10^{-6}$; 22.0-fold enriched); response to organic substance (open yellow circle, $p = 1.05 \times 10^{-5}$; 6.4-fold enriched); and lipid homeostasis (yellow dot, $p = 6.93 \times 10^{-5}$; 36.2-fold enriched).

The following source data and figure supplements are available for figure 8:

**Source data 1**. Table of DAVID enrichments used to annotate *Figure 8* and the HGMD genes that overlapped our CRMs and singletons in the different phylogenetic categories.

**Figure supplement 1**. CRMs link TF binding with disease-causing variants in blood coagulation.

---

modifications) as classifiers. We compared the 84,301 human developmental enhancers predicted by EnhancerFinder to our data. While all phylogenetic categories overlapped significantly (*Heger et al., 2013*), the Deeply shared CRMs gave the highest overlap (~30%) with the EnhancerFinder predictions. The evidence we present here for the functional relevance of Deeply shared combinatorial TF–DNA interactions in primary liver tissue suggests that results from such empirical studies in primary tissues will be another valuable source of information which can be utilized in integrative methods for enhancer prediction.

Just as using DNA constraint as a sole criteria for enhancer finding has its limitations, using filtering approaches that require conserved combinatorial binding will also miss important regulatory events

near genes of interest. For example, the common SNP rs2279744 (also known as 'SNP 309') is found in the first intron of the *MDM2* and contributes to carcinogenesis in humans by increasing levels of MDM2, a negative regulator of TP53 (*Bond et al., 2004*). This SNP has been shown to enhance the binding of the SP1 transcription factor at a site that does not have a clear orthologous mouse motif. For this reason, transgenic mice with human intron 1 alleles have been created and show a cancer phenotype (*Post et al., 2010*). This example illustrates where strictly using sequence conservation alone would fail to identify a functional regulatory variant. Interestingly, in our data we found a Deeply shared human HNF4A singleton TF binding event spanning rs2279744. Although the role of HNF4A at the MDM2 intron 1 has yet to be characterized, rs2279744 also serves as an example where excluding a presumptive singleton TF binding events from functional analyses would have missed an important regulatory variant.

Our results suggest that conserved liver regulatory regions reside near SNPs or rare mutations associated with liver related phenotypes. The most striking disease associations we observed involved the blood coagulation pathway. This pathway is perhaps one of the best-studied biological pathways and has long served as a model system for understanding human disease gene mutations. This rich history allowed us to observe, perhaps for the first time, how recurrent TF binding mutations found within conserved combinatorially occupied TF binding sites can be afflicted on several members of the same pathway. Conserved combinatorial control regions were also found near proteins that are part of the coagulation and complement system (e.g., C1R, C1S, C2, C4B, C4B, C4BPA, C4BPB, C5, C6, C7, C8A, C8B, C8G, C9, CFB, CFH, and CPB2). The interplay between the coagulation and complement system proteins has long been appreciated (reviewed by *Markiewski et al., 2007*), and our results suggest that their gene expression in the liver is coordinated by conserved cis-regulatory modules.

Overall, the observation of recurrent phenotype-causing regulatory mutations in a single pathway is likely a phenomenon that occurs in other tissues and biological pathways. Our study suggests that identifying sites of shared combinatorial binding will be relevant criteria for assigning pathological significance to candidate disease variants uncovered in whole genome sequencing studies.

## Materials and methods

### Molecular biology and genomics

#### Tissue preparation

Tissues from all five species were treated post-mortem with 1% formaldehyde as previously described (*Schmidt et al., 2009*). The dog liver material (Cfam; 2 adult males; 14 months of age) was obtained from commercial sources (Harlan). Human liver material (Hsap, 2 males, unknown age) was obtained from the Liver Tissue Distribution Program (NIDDK Contract #N01-DK-9-2310) at the University of Pittsburgh and from Addenbrooke's Hospital, Cambridge under the human tissue license (08/H0308/117). Mmus (adult C57BL6/J males, 2.5 months of age) was obtained from the CRI under Home Office license PPL 80/2197. Macaque (Mmul) material was purchased from the CFM, UK.

#### ChIP-Sequencing

The CEBPA antibody sc-9314 (Santa Cruz Biotech, CA) and HNF4a antibody ARP31946 (Aviva Biosystems, CA) were used as previously described (*Schmidt et al., 2010*). FOXA1 antibody ab5089 (Abcam, UK) has been previously used for FOXA1 ChIP–chip and ChIP-seq experiments in human and mouse (*Motallebipour et al., 2009*; *Hurtado et al., 2011*). ONECUT1 (HNF6) anti-human polyclonal antibody sc-13050 (Santa Cruz Biotech) has previously been described for human and mouse ChIP–chip (*Odom et al., 2007*; *Wilson et al., 2008*). Further details on the antigens used to raise the antibodies used for ChIP-seq are shown in *Figure 1—source data 1D*. Briefly, the immunoprecipitated material was end-repaired, A-tailed, ligated to single- or paired-end sequencing adapters, amplified by 18-cycles of PCR and size selected (200–300 bp) followed by single end sequencing on an Illumina Genome Analyzer II according to the manufacturer's instructions.

#### Data

ChIP-seq data generated for this study includes: CEBPA, FOXA1, ONECUT1, and HNF4A in rhesus macaque, FOXA1 in human, and FOXA1 and ONECUT1 in dog. Experiments used in this study that

have been previously reported by our team include: ChIP-seq experiments for CEBPA and HNF4A for human and dog (*Schmidt et al., 2010*; E-TABM-722); ChIP-seq for Hnf4a, Cebpa, and Foxa1 in C57BL6/J mice (E-MTAB-1414), as well as for Brown Norway rat (E-MTAB-1415) (*Stefflova et al., 2013*); Human and mouse ONECUT1 ChIP-seq (E-MTAB-890) (*Funnell et al., 2013*); and RNAseq data for all five species (*Kutter et al., 2011*; E-MTAB-424). To facilitate analysis of this data set by others, all ChIP-seq files used here have been deposited in ArrayExpress under a single accession number (E-MTAB-1509). We filled genomic gaps in the coding regions for dog FOXA1 and macaque ONECUT1 and deposited them under NCBI accessions numbers JN601139 and JQ178331 respectively. Illumina BodyMap data were obtained from E-MTAB-513.

## Computational biology and analysis

### Sequence alignment and peak calling

ChIP-seq and input reads from each species were aligned with MAQ (*Li et al., 2008*) using default parameters to their respective genome assemblies (human [NCBI 36], macaque [Mmul_1], mouse NCBI m37; rat [RGSC3.4] and dog [CanFam2.0]). All sequence, genome annotations, and comparative genomics data were taken from Ensembl release 52.

For each TF, at least two biological replicates were performed and aligned to their respective genomes (*Figure 1—source data 1A–C*) (*Landt et al., 2012*). Sequence reads from high quality replicates were pooled prior to calling peaks used for all comparative analyses and building CRMs. Peak calling was performed using SWEMBL with parameters '-R 0.005 -i –S' as described previously (http//www.ebi. ac.uk/~swilder/SWEMBL/; *Schmidt et al., 2012*). The correlation of normalized read counts between biological replicates in the ChIP-seq data was further assessed using the DiffBind Bioconductor package (*Stark and Brown, 2011*) (*Figure 1—source data 1C*). ChIP-seq data was also assessed using the quality control standards proposed by the ENCODE consortium and further described by *Marinov et al. (2014)*. In the case where the second replicate could validate the first replicate, but gave substantially less peaks, we only used the one replicate that gave the most ChIP-seq binding events for our downstream comparative analyses. This occurred for: macaque CEBPA, macaque HNF4A, macaque FOXA1, and dog ONECUT1. Although the human female ONECUT1 was a good replicate, we only used peaks from the human male ONECUT1 experiment to confine our comparative analysis to one sex.

### Motif discovery

Motif discovery was conducted with MEME (*Bailey et al., 2009*) using the default settings and the following parameters (-revcomp -maxw 20 -minw 6 -nmotifs 5). We selected the top 500 peaks ordered by input-corrected read depth and used the 25 bp centered on the identified summit of the peak as our input for the motif discovery analysis. As we previously reported for CEBPA and HNF4A, the position weight matrices (PWM) for FOXA1 and ONECUT1 were nearly identical between species (*Figure 1—figure supplement 1*) (*Schmidt et al., 2010*). For *Figure 8—figure supplement 1*, we used the mouse PWMs for each TF to identify motif location within all of our CRMs using the programs PERL TFBS (*Lenhard and Wasserman, 2002*) and CENTIPEDE (*Pique-Regi et al., 2011*). This motif location information was used in conjunction with ChIP-seq signal to manually annotate promoters and regulatory variants (*Figure 8—figure supplement 1*).

Search for secondary motifs was done using *peak-motifs* from RSAT (*Thomas-Chollier et al., 2011*) with default parameters. *peak-motifs* uses a word count approach which enables users to input large sequence sets and recover motifs in a short time, in contrast with most alignment based approaches like MEME. CRMs and Singleton binding regions were classified in accordance to their conservation status, and these separated sets were used as input. Output motifs were compared with motifs discovered using MEME (see above) and external motifs coming from JASPAR (Vertebrate core 2009 version) (*Sandelin et al., 2004*) and UniPROBE (*Newburger and Bulyk, 2009*) (2009 version) databases, in order to assign identity to the discovered motifs.

Motif comparison was performed using *compare-matrices* from RSAT. The program calculates the correlation between two matrices shifting positions; the correlation is normalized based on the width of the alignment to avoid high correlation based on few flanking positions (*Supplementary file 1*).

### Cis-regulatory module (CRM) construction

We first computed the distances separating any two TF summits along the genome. ChIP-seq peaks were clustered into CRMs when the peak summits from at least two distinct TFs (e.g., CEBPA and

HNF4A) fell within 300 bp (see *Figure 1* for graphical explanation). (*Figure 1A*, *Figure 1—figure supplement 3*). All other TFs were treated as singletons.

## Interspecies overlap of transcription factor binding and CRMs

Shared binding events and CRMs were identified using the 9-way Enredo-Pecan-Ortheus (EPO; *Paten et al., 2008*) multiple sequence alignments from the Ensembl Compara database (*Flicek et al., 2010*). We performed these comparative analyses requiring different overlaps within the MSA criteria (1 bp, 10 bp, 25 bp, and 50 bp) (*Figure 1—source data 1G*). This range of parameters did not overtly affect the number of shared CRMs we retrieved. For the results presented, we define a CRM (or single TF) occupied region to be shared if it overlaps at least 10 bp of a CRM (or single TF) occupied region in a second species. See *Figure 2—source data 1* for all the CRMs and singletons along with their phylogenetic categories.

## Liver RNA-seq data analysis

Liver RNA-seq data from the same species were used to analyze if CRM status and conservation were related to gene expression levels. Polyadenylated mRNA levels were assessed using previously described RNA-seq data (E-MTAB-424; *Kutter et al., 2011*). Briefly, trimmed 36 bp reads were aligned using TopHat and FPKM estimates were obtained using Cufflinks (*Trapnell et al., 2012*) for all features annotated in Ensembl 52. Individual TF binding events and CRMs were associated to the nearest gene. The number of shared species for individual TF binding events and CRMs were recorded and associated with expression data.

## Illumina Body Map analysis

We retrieved raw FASTQ files from ArrayExpress (accession E-MTAB-513). We first filtered these data with prinseq-lite (v0.15) and fastx (v0.0.13): we trimmed both read ends with a quality threshold of 20, and we then discarded those reads that were shorter than 35 nucleotides, had more than 5% of Ns, a dust score bigger than 10 (i.e., low complexity reads), and more than 5% of the bases with a quality inferior to 30. This subset of high quality reads was then mapped to the genome using BWA (v0.5.9, default options). We retrieved mapped reads and filtered them further by removing reads with a mapping quality below 20 (including multireads). In the case of paired-end data, we also removed all reads that were not properly paired. Finally, we calculated exon expression levels using htseq-count (HTSeq v0.5.3p3) and averaged those to produce gene expression estimates. TF binding sites not found in CRMs (Singletons) or within CRMs (Modules) were associated with the nearest gene. The FPKM (Fragments Per Kilobase of exon per Million reads) for each gene was recorded (obtained from the Human BodyMap 2.0 dataset from Illumina) and displayed as a heatmap (*Figure 5*).

## Comparison to ENCODE data

We first compiled data relevant to transcriptional regulation from ENCODE, for all cell lines. For each ENCODE data set, we then calculated its observed overlap with each of our ChIP data sets (i.e., 'Deeply shared CRMs', and 'Shared Singletons') by counting the number of regions that overlap by at least 1 bp. We then built a distribution of expected overlap values from 1000 iterations of randomizing the input regions, by shuffling each one into a random genomic region chosen from among the DNaseI hypersensitive regions in any cell type (i.e., only shuffling them into possible regions of open chromatin, as a conservative estimate of possible regions where TFs might bind). The distribution of the overlap scores from the randomized data resembles a normal distribution, which we used to generate a Z-score and p-value for the observed number of peaks that overlap each real data set (see *Supplementary file 1*).

## Functional enrichment analyses using GREAT

Functional enrichment of ChIP-seq data was performed using the online tool GREAT version 2.0.2 (*McLean et al., 2010*). Importantly, GREAT is designed for ChIP-seq data analysis unlike hypergeometric enrichment analyses used for gene expression data. Hg18 assembly was used with the Association Rule: Basal+extension: 5 kb upstream, 1 kb downstream, 1000 kb max extension (*Figure 3*, *Figure 6*). GREAT figures were produced with an in-house R tool and are ranked according to the same default criteria used by the GREAT (binomial FDR q-value is ≤0.05 with a significant hypergeometric FDR q-value and a minimum region fold enrichment of 2). The −log10 binomial FDR q-values were plotted on the x-axis. Q-values for up to a maximum of five of the most significant enrichment categories obtained for each test data set are shown. Significant enrichments are labeled with an asterisk, which is drawn proportional to the fold enrichment value (*Figure 3—source data 1*).

## Overlap of shared CRMs with curated human regulatory mutations

Human CRMs that had TF binding in syntenic regions in at least two additional species (n = 5046) were intersected with the Human Gene Mutation Database (HGMD; professional version). A representative network analysis of these genes was obtained using the GeneMANIA server using default parameters (*Warde-Farley et al., 2010*). (*Figure 8*, *Figure 8—source data 1*). Using this data and the Default parameters for its Pathway analysis, GeneMania drew upon information from six sources (five of which are within Pathway Commons) and the sixth being *Wu et al. (2010)*. For its physical interaction analysis, it utilized 190 data sources supported by five data sets from iRefIndex (*Razick et al., 2008*) and one study in BioGrid (*Zanon et al., 2013*). Functional annotation enrichment of these genes was performed using DAVID (*Huang da et al., 2009*).

## Location and GERP analysis of CRMs and singletons

To define the locations of each regulatory region, we sliced the genome into categories: TSS ±3 kb, intronic, intergenic, and exonic. We obtained all transcripts (exons and introns) defined by the Ensembl genebuild, keeping the longest features. Any regions falling within ±3 kb of a TSS was characterized as the TSS ±3 kb category. In the event of complex gene arrangements or regions bridging two categories, we prioritized the locations as follows: TSS ± 3 kb > intron > intergenic > exon. Conservation scores were obtained from the Ensembl Compara database (*Flicek et al., 2010*). Genomic Evolutionary Rate Profiling (GERP; *Cooper et al., 2005*) score was used to calculate the conservation of each nucleotide in multi-species alignment The multiple alignment used to derive GERP score is the 29-way EPO alignment. For CRMs and singletons, GERP scores were extracted for each base pair ±1 kb of the center of the CRMs or the peak summit for singletons.

## Intersecting GWAS with shared CRMs

Linked SNPs and $r^2$ values were obtained using the Ensembl Variation Perl API (*Rios et al., 2010*) with the 1000 Genomes Project pilot 1 (*1000 Genomes Project Consortium et al., 2010*) low coverage CEU panel genotypes. Linkage disequilibrium measures were calculated between target SNPs from the NHGRI GWAS catalog (*Hindorff et al., 2009*) up to a maximum of 100 kb upstream and downstream of the lead SNP. Prior to overlapping with CRMs/singletons, windows around each SNP were calculated based on the largest genomic interval between two SNPs in linkage disequilibrium with the lead SNP ($r^2 \geq 0.8$). Alternatively, each GWAS lead SNP was given a window of ±2.5 kb. The GWAS catalog (25 August 2011) was manually summarized into 25 categories using the given Disease Trait, Experimental Factor Ontologies (EFO; *Malone et al., 2010*), and the primary literature. The motivation for this classification was to group GWAS studies that had a clear tissue-specificity (e.g., liver, immune system) or disease type (e.g., diabetes, cancer) (see *Figure 7—figure supplement 1* for overview of classification and *Figure 7—source data 1H* for the disease trait/summary key used). Lead SNPs whose window overlapped were collapsed into a single interval. Identical disease trait/summaries for lead SNPs within the same interval were counted only once.

We implemented two null models to further test the significance of the enrichments we found. First, we shuffled the GWAS SNP summary annotation 1000 times. Using each random set, we annotated CRMs and singletons with the summaries assigned to overlapping SNPs as described for *Figure 7*. We then calculated the enrichment for each summary term in the various phylogenetic categories. Second, we created a random set of SNPs from the full list of GWAS lead SNPs that matched the minor allele frequency and distance to transcription start site that we observed in our analyses in *Figure 7*. This was repeated 1000 times to create a null distribution of p-values. We considered an enrichment significant if less than 5% of the 1000 p-values obtained from either of the two null models had an enrichment p-value equal or lower to the one obtained using the original GWAS SNPs summary annotation sample before multiple testing correction.

Analyses were also performed by collapsing the NHGRI Disease Traits (see *Table 1*). All hypergeometric tests and heatmap plots were performed using R and were corrected for multiple testing.

## Acknowledgements

We thank the Cancer Research UK—Cambridge Institute cores: genomics (James Hadfield) and bioinformatics (Matthew Eldridge) and the BRU (Selina Ballantyne) for support. We thank Vinh Truong for help with the GWAS null model implementation. We also acknowledge Dani Welter, Fiona Cunningham, Sushmita Roy, and Alan Moses for helpful discussions.

## Additional information

### Funding

| Funder | Grant reference number | Author |
|---|---|---|
| European Research Council | 202218 | Duncan T Odom |
| EMBO | Young Investigator Award | Duncan T Odom |
| SickKids Foundation | | Alejandra Medina-Rivera, Matthew Carlucci, Kyle Chessman, Michael D Wilson |
| Natural Sciences and Engineering Research Council of Canada | 436194-2013 | Michael D Wilson |
| Wellcome Trust | WT098051 | Duncan T Odom, Paul Flicek |
| INSERM | | Benoit Ballester |
| Canada Research Chairs | | Michael D Wilson |
| Marie Curie Reintegration Grant | | Klara Stefflova |
| Swiss National Science Foundation | | Claudia Kutter |
| Cancer Research UK | | Michael D Wilson, Dominic Schmidt, Claudia Kutter, Margus Lukk, Suraj Menon, Klara Stefflova, Stephen Watt, Duncan T Odom |
| Consejo Nacional de Ciencia y Tecnología | | Alejandra Medina-Rivera |
| European Molecular Biology Laboratory (EMBL) | | Benoit Ballester, Mar Gonzàlez-Porta, Andre J Faure, Angela Goncalves, John C Marioni, Paul Flicek |
| Wellcome Trust | WT095908 | Benoit Ballester, William M McLaren, Paul Flicek |
| European Molecular Biology Laboratory | International PhD Progam | Mar Gonzàlez-Porta, Andre J Faure, Angela Goncalves |
| Heart and Stroke Foundation of Ontario | Bridge grant:7486 | Alejandra Medina-Rivera, Michael D Wilson |

The funders had no role in study design, data collection and interpretation, or the decision to submit the work for publication.

### Author contributions

BB, PF, Conception and design, Analysis and interpretation of data, Drafting or revising the article; AM-R, MG-P, MC, XC, KC, AJF, APWF, AG, ML, SM, WMML, MC, JCM, Analysis and interpretation of data, Drafting or revising the article; DS, Conception and design, Acquisition of data, Drafting or revising the article; CK, KS, SW, Acquisition of data, Drafting or revising the article; MTW, Acquisition of data, Analysis and interpretation of data, Drafting or revising the article; DTO, MDW, Conception and design, Acquisition of data, Analysis and interpretation of data, Drafting or revising the article

### Ethics

Human subjects: Human liver material (Hsap, 2 males, unknown age) was obtained from the Liver Tissue Distribution Program (NIDDK Contract #N01-DK-9-2310) at the University of Pittsburgh and

from Addenbrooke's Hospital, Cambridge under the human tissue license (08/H0308/117). Mmus (adult C57BL6/J males, 2.5 months of age) were obtained from the CRI under Home Office license PPL 80/2197. Animal experimentation: Mmus (adult C57BL6/J males, 2.5 months of age) were obtained from the Cancer Research UK-Cambridge Institute under Home Office license PPL 80/2197.

## Additional files

### Supplementary files

• Supplementary file 1. Motif enrichments of CRMs and singletons as well as overlaps of CRMs and Singletons with ENCODE data.

• Supplementary file 2. Deeply conserved CRMs within 2.5 kb of a lead GWAS SNP were overlapped with RegulomeDB annotated SNPs. SNPs with a high confidence evidence of regulatory potenital are shown (evidence levels 1 and 2 from Regulome DB).

• Supplementary file 3. Tally of genes with HGMD regulatory mutations that overlap liver CRMs or singletons.

### Major datasets

The following datasets were generated:

| Author(s) | Year | Dataset title | Dataset ID and/or URL | Database, license, and accessibility information |
|---|---|---|---|---|
| Wilson MD, Ballester B, Schmidt D, Kutter C, Lukk M, Stefflova K, Watt S, Flicek P, Odom DT | 2014 | ChIP-seq of liver samples from five placental mammals to study the evolution of combinatorial transcription factor binding | E-MTAB-1509; http://www.ebi.ac.uk/arrayexpress/ | Publicly available at EBI ArrayExpress. |
| Wilson MD, Watt S, Odom DT | 2012 | Canis lupus familiaris FoxA1 (FoxA1) gene, partial cds | http://www.ncbi.nlm.nih.gov/nuccore/JN601139 | Publicly available at NCBI. |
| Wilson MD, Watt S, Odom DT | 2012 | Macaca mulatta Onecut1 protein (Hnf6) gene, partial cds | http://www.ncbi.nlm.nih.gov/nuccore/JQ178331 | Publicly available at NCBI. |

The following previously published datasets were used:

| Author(s) | Year | Dataset title | Dataset ID and/or URL | Database, license, and accessibility information |
|---|---|---|---|---|
| Schmidt D, Wilson MD, Ballester B, Schwalie PC, Brown GD, Marshall A, Kutter C, Watt S, Martinez-Jimenez CP, Mackay S, Talianidis I, Flicek P, Odom DT | 2010 | ChIP-seq of Canis familiaris, Gallus gallus, Mus musculus, Homo sapiens, Monodelphis domestica to investigate CEBPA and HNF4a binding in five vertebrates | E-TABM-722; http://www.ebi.ac.uk/arrayexpress/experiments/E-TABM-722/ | Publicly available at EBI ArrayExpress. |
| Schroth GP | 2011 | RNA-Seq of human individual tissues and mixture of 16 tissues (Illumina Body Map) | E-MTAB-513; http://www.ebi.ac.uk/arrayexpress/experiments/E-MTAB-513/ | Publicly available at EBI ArrayExpress. |
| Kutter C, Brown GD, Watt S, Wilson MD, Goncalves A, White RJ, Odom DT | 2011 | RNA-Seq of six mammals | E-MTAB-424; http://www.ebi.ac.uk/arrayexpress/experiments/E-MTAB-424/ | Publicly available at EBI ArrayExpress. |
| Funnell APW, Wilson MD, Ballester B, Mak KS, Burdach JG, Magan N, Pearson RCM, Lemaigre F, Stowell KM, Odom DT, Flicek P, Crossley M | 2013 | A CpG mutational hotspot in a ONECUT (HNF6) binding site accounts for the prevalent variant of Hemophilia B Leyden | E-MTAB-890; http://www.ebi.ac.uk/arrayexpress/experiments/E-MTAB-890/ | Publicly available at EBI ArrayExpress. |

| Stefflova K, Thybert D, Wilson MD, Streeter I, Aleksic J, Karagianni P, Brazma A, Adams DJ, Talianidis I, Marioni JC, Flicek P, Odom DT | 2013 | ChIP-seq of mouse adult liver cells to investigate transcription factor binding region between five species | E-MTAB-1414; http://www.ebi.ac.uk/arrayexpress/experiments/E-MTAB-1414/ | Publicly available at EBI ArrayExpress. |
| Stefflova K, Thybert D, Wilson MD, Streeter I, Aleksic J, Karagianni P, Brazma A, Adams DJ, Talianidis I, Marioni JC, Flicek P, Odom DT | 2013 | ChIP-seq of adult wild-type rat liver cells to study genome wide location of HNF4a, FoxA1 CEBPA | E-MTAB-1415; http://www.ebi.ac.uk/arrayexpress/experiments/E-MTAB-1415/ | Publicly available at EBI ArrayExpress. |

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
