## [Decision Letter]

Thank you for sending your work entitled “Multi-species, multi-transcription factor binding highlights conserved control of tissue-specific biological pathways” for consideration at *eLife.* Your article has been favorably evaluated by Stylianos Antonarakis (Senior editor) and 2 reviewers (Drs Mike Beer and Ross Hardison).

The Senior editor and the reviewers discussed their comments before we reached this decision, and the Senior editor has assembled the following comments to help you prepare a revised submission.

There are several major points that need to be addressed in the revised document (please see below). In addition both reviewers have suggested a reorganization of the display items.

*Reviewer 1*:

1) A main result is that most CRMs “are evolving rapidly”, but to be precise, what “conserved” means here is that the sequences are not “alignable by EPO.” This is naively in conflict with Figure 1 showing that TFBS are conserved in all species. It seems that the authors have the ability to investigate what is evolving, either: 1) CRM TFBS composition (binding sites evolving in our out), 2) TFBS linear arrangement (binding sites reordered), or 3) CRM-gene association (entire CRMs relocating and losing synteny). Another possibility is that all 3 of these features are conserved, but what is called lack of conservation is simply a failure of EPO alignment to detect the conserved TFBS arrangement and composition. There are computational approaches specifically designed for this case (where PWMs and TFBS are known) which outperform EPO for this task (compared in Su Teichman Down 2010) and something along the lines of Kim, He, Sinha 2009; Sinha, He 2007; He, Ling, Sinha 2010; all in PLOS Comp Bio, should be utilized.

2) Details of CRM construction are too sparse, and it is not very clear what CRM composition is until Supplemental Table 2, which should be in Figure 1, and numbers of singletons should be added, not buried in the supplementary material. If a CRM changes its TFBS composition across species, but the sequence is alignable, is it still considered conserved?

3) How sensitive are the results to the specific cutoffs used for ChIP-seq signal to quantify a binding event vs. non-event? The analysis should be performed on different replicates to test this. How many CRMs change their composition using only one repl to build them vs. the other? Also, it seems a bit inconsistent to say that the ENCODE quality control standards are used for the replicates if later only the replicate which gives the most peaks is used for downstream analysis. If both replicates don't give similar results, how robust are the conclusions?

4) Motif detection is good, but enrichment should be quantified. The fraction of CRM sequences which have motifs present above some cutoff PWM score for each of the 4 PWMs should be reported for all CRMs and singletons. E.g., how many singleton HNF4A binding regions have a PWM instance above some cutoff, compared to how many singleton sites for the other 3 factors have a HNF4A PWM score above that same cutoff?

Reviewer 2:

The current manuscript is difficult to read and understand. I have several recommendations for revisions, starting with some major re-organization.

1) The logical flow is fairly strong in the Results, but the data referred to are dispersed across many pages of Figures and Figure supplements. Also the order of the main figures does not fit with the presentation in the Results: it really is not used until the third section. I strongly recommend re-organizing the figures. Start with some of the current Figure 1 supplements, specifically Figure 1—figure supplement 3, 4, 5 to show the basic data, and how you constructed the CRMs, and the amount of conservation for TF bound sites and CRMs. Then go to the current Figure 1, which describes how the CRMs are categorized by extent of conservation.

2) Figure 2 and the many supplemental figures have multiple problems. The authors have developed an excellent way of displaying the GREAT results, but they flood the reader with too much information. A better-digested and focused set of key results needs to be featured. The text does have a good focus; however, the text and figures are not closely aligned. The specific examples of significant enrichments in the text are not listed in the main Figure 2, and the reader has to scan through four or five pages of figure supplements to find the data being discussed. One figure highlighting the key enrichments needs to be refined from all this material. Also, the method for assigning likely target genes is not stated in the main text. This is a very important point for interpreting the data, and it should be stated in the Results as well as in the Methods.

3) The description and interpretation of the data in Figure 4 need to be more accurate. The data in panel B are over-interpreted in the legend and in the Results. The expression of presumptive targets is not limited to liver (“uniquely expressed”) but the expression levels are higher there. The text says “Unlike CRMs, conserved singletons do not show a strong increase in liver-specific gene expression.” In fact, presumptive targets of singletons show a similar pattern to the presumptive targets of CRMs, but the levels of expression are lower. Is the decrease in signal for deeply shared singletons significant, or is this related to a small number of targets to examine? Also, the legend should specify the RNA-seq data used in panel A. The top comparison in Figure 4 seems to be between the expression levels of presumptive targets for deeply conserved CRMs and lineage-specific singletons, but it has a surprisingly high p-value; this should be clarified.

4) Figure 6 main and Figure 6–figure supplement 1: The authors are making a critical point but the connections between the text and the Figures are not stated clearly. In the Results, the authors give p-values for enrichment of about 0.005 to 0.008 and refer to Figure 6. However, those p-values are MUCH smaller, and correspond to the analysis in the next sentence, after separation by conservation categories. However, the reader is referred to Supplemental File 4, whereas they really need to look at Figure 6! The fact that the enrichments are robust to different ways of dealing with potentially functional SNPs in proximity or LD is a strong point and accurately stated in the Results. However, the titles to the figure legends (Figure 6 vs. Figure 6–figure supplement 1) are the same, and it takes some careful reading to figure out what is different about the two figures.

5) The Discussion should address what is missed by focusing on only combinatorial binding, only conserved binding, and both. Clearly, many fewer bound sites are examined by doing this filtering, but the enrichment for likely function is substantial. What are the potential costs of taking this approach?

6) I would like to see a paragraph in the Discussion comparing the advantages of filtering vs integration. Are there integrative approaches that would emphasize the features shown to be useful here (combinatorial binding and conservation) while not completely excluding other binding sites?

7) What is the null model for the tests of enrichment for GWAS SNPs in CRMs, singletons (Figure 6)? In previous studies, the null model was restricted to SNPs on the genotyping array that were not associated with a trait but were matched to the trait-associated SNPs in several important ways, e.g. allele frequency, position relative to gene structure, and proximity to the transcription start site.

---

## [Author Response]

Reviewer 1:

*1) A main result is that most CRMs “are evolving rapidly”, but to be precise, what “conserved” means here is that the sequences are not “alignable by EPO.” This is naively in conflict with*
Figure 1
*showing that TFBS are conserved in all species. It seems that the authors have the ability to investigate what is evolving, either: 1) CRM TFBS composition (binding sites evolving in our out), 2) TFBS linear arrangement (binding sites reordered), or 3) CRM-gene association (entire CRMs relocating and losing synteny). Another possibility is that all 3 of these features are conserved, but what is called lack of conservation is simply a failure of EPO alignment to detect the conserved TFBS arrangement and composition. There are computational approaches specifically designed for this case (where PWMs and TFBS are known) which outperform EPO for this task (compared in Su Teichman Down 2010) and something along the lines of Kim, He, Sinha 2009; Sinha, He 2007; He, Ling, Sinha 2010; all in PLOS Comp Bio, should be utilized*.

We thank the reviewer for pointing out the ambiguity on what we define as being conserved in this study. This is relevant to several other comments below. We have now replaced the term “conserved” with “shared” to make it clear that CRMs or Singletons that overlapped in the EPO multiple sequence alignment are what we are focusing on. This overlap in the EPO-MSA did not require more than 10 bp and was not based on CRM content or motif alignment. To better make this point, and to address reviewer 2’s comment about organization of the manuscript, we created a new Figure 1 to illustrate our method and terminology. We also indicate the fraction of human CRMs and Singletons of the different phylogenetic categories that fall within the EPO alignment of each of the other four species.

Using five mouse species/strains we recently described the evolution of CRM content and turnover (see [72]). Based on the virtue of high quality multiple sequence alignments, and the controlled nature of tissue acquisition and biological variables that working with rodents affords, we believe that the results from [72] provide a detailed answer to this important question. Here we did not attempt to track the gain and loss of individual TF binding within a CRM but rather we tried to use a “lenient” means of assembling CRMs (any two TFs) with the goal of capturing as many shared CRMs as possible for functional analysis.

Our new Figure 1 shows that most (>93%) of human CRMs and singletons we detect are found in EPO with macaque, which suggests that the rapid turnover observed between human and macaque CRMs is not due to characteristics of the multiple alignment. Greater than 85% CRMs or singletons that are shared in at least one additional non-primate (Beyond primate category) are aligned within a non-primate species. This indicates that again the loss of a CRM in one or more species is unlikely due to the absence of the sequence in EPO.

While current methods which use only sequence properties will not necessarily be able to distinguish liver CRMs from all regulatory regions, these computational methods are highly relevant to compare to our dataset. We looked into the above papers and the methods reviewed in Su et al. 2010. From this paper the “Regulatory Potential Score” appeared to be the most promising and accessible regulatory prediction dataset we could compare to our data. Using the GAT tool [29], we asked which of our phylogenetic categories were captured by the “ESPERR Regulatory Potential (7 species) described by Taylor et al. 2006 and Kolbe et al. 2004. As expected, of all of the phylogenetic categories significantly overlapped ESPERR regions, with the deeply shared CRMs having the highest overlap. ∼16% of the deeply shared CRMs within the deeply share CRMs overlapped ESPERR regions.

We also compared our data to a recent paper, which employed a novel method that successfully integrates a variety of functional data and sequence properties to predict developmental enhancers (15). This method was trained on experimentally verified human VISTA enhancers and utilizes evolutionary conservation, DNA motifs and functional genomics data (such as p300 and histone modifications) as classifiers. EnhancerFinder clearly out performs individual classifiers for predicting enhancers. We compared the 84,301 predicted human developmental enhancers predicted by EnhancerFinder to our data. Again, as expected, all categories significantly overlapped the EnhancerFinder data, with the deeply shared CRMs giving the highest overlap. ∼30% of deeply shared CRMs overlap EnhancerFinder predictions. These results suggest that studies of combinatorial TF binding in primary tissues from multiple species will provide valuable datasets that are currently missed bin current integrative analyses. This result has been mentioned in the Discussion and incorporated into the response to Reviewer 2 comment 6.

*2) Details of CRM construction are too sparse, and it is not very clear what CRM composition is until Supplemental*
Table 2*, which should be in*
Figure 1*, and numbers of singletons should be added, not buried in the supplementary material*. *If a CRM changes its TFBS composition across species, but the sequence is alignable, is it still considered conserved?*

This is a crucial comment raised also by reviewer 2. We have now provided a new Figure (Figure 1) illustrating the method for CRM construction. If a CRM changes TFBS composition and still remains a CRM, we consider it shared. We have also now consistently referred to the overlap of CRMs/singletons in our data as being “shared” rather than “conserved”.

We also clarified the legends on the CRM building in what is now Figure 1—figure supplement 3. We moved the number of peaks and singletons out of the [Supplementary-material SD6-data] and into called to the main Figure 1—figure supplement 1. We also added a [Supplementary-material SD2-data] which has all the CRMs and singletons along with their phylogenetic categories listed.

*3) How sensitive are the results to the specific cutoffs used for ChIP-seq signal to quantify a binding event vs. non-event? The analysis should be performed on different replicates to test this*. *How many CRMs change their composition using only one repl to build them vs. the other? Also, it seems a bit inconsistent to say that the ENCODE quality control standards are used for the replicates if later only the replicate which gives the most peaks is used for downstream analysis. If both replicates don't give similar results, how robust are the conclusions?*

We asked how building human CRMs with: 1) individual ChIP-seq biological replicates or 2) using a different peak caller (MACS), affected the phylogenetic classification of CRM/Singletons. We built human CRMs with the replicate that gave the most peaks “Hi rep” and with the replicate that gave the fewest peaks “Lo rep”. We also used the MACs peak caller at an FDR of 0.05, which is more permissive at calling peaks than SWEMBL. The results for this comparison is tabulated and plotted in [Supplementary-material SD6-data]. This comparison shows: overall the shared CRMs were robust to replicate and peak caller differences; few CRMs classified in this study became singletons using different replicates or peak caller; the majority of new peaks called fell into human only singletons; when a newly called CRM was classified as a singleton TF binding event in our study, the majority of these new CRMs were shared only in human. Finally when more peaks were called (e.g., using MACs or our “Hi rep” replicate ChIP-seq data), approximately 40% of the shared human singletons identified in our study became classified as CRMs. Although it is not unexpected that the singleton TF binding category is more unstable than the CRM category, our comparative analysis of shared singleton TF binding category has yielded novel genomic features (e.g. proximity to the TSS, high DNA constraint, and a relative reduction in liver-specific enrichments) which warrant future exploration. We have now raised this point in our Results and included this analysis in [Supplementary-material SD1-data].

[44] recently performed the ENCODE QC of [38] on all the public ChIP-seq data in the NCBI GEO database. As our previously published (and of course unpublished) primary data is in ArrayExpress, our multi-species ChIP-seq data had not been subject to this QC. We felt that having this QC performed on our data would allow others who wish to use our data to quickly get a sense of its quality. In Landt et al, they do suggest cases where one replicate is used for a study and suggest that such cases, if they occur, should be explained. We now cite [44] in this manuscript to justify our use of ENCODE quality control measures. We hope that addressing the reviewers comments regarding the CRM quality have made it clear that we have done our best to assemble and compare CRMs. We have raised this issue in the discussion as part of the answer to Reviewer 2 comment 5 and in the singleton versus CRM results section.

*4) Motif detection is good, but enrichment should be quantified. The fraction of CRM sequences which have motifs present above some cutoff PWM score for each of the 4 PWMs should be reported for all CRMs and singletons*. *E.g., how many singleton HNF4A binding regions have a PWM instance above some cutoff, compared to how many singleton sites for the other 3 factors have a HNF4A PWM score above that same cutoff?*

We thank the reviewer for this suggestion. We performed this analysis and have incorporated the results into the paper (text starting “We asked whether CRMs or singletons differ with regards to the quality of their TF binding motifs. Peaks for each TF were scanned using the RSAT tool matrix-scan with the best position weight matrices (PWM) for each TF”). This allowed us to highlight that singleton TF binding events have a great fraction of high quality motifs compared to CRMs.

Reviewer 2:

*The current manuscript is difficult to read and understand. I have several recommendations for revisions, starting with some major re-organization*.

*1) The logical flow is fairly strong in the Results, but the data referred to are dispersed across many pages of Figures and Figure supplements. Also the order of the main figures does not fit with the presentation in the Results: it really is not used until the third section. I strongly recommend re-organizing the figures. Start with some of the current Figure 1 supplements, specifically*
Figure 1—figure supplement 3*, 4, 5 to show the basic data, and how you constructed the CRMs, and the amount of conservation for TF bound sites and CRMs. Then go to the current*
Figure 1*, which describes how the CRMs are categorized by extent of conservation*.

We have taken the reviewer’s suggestion and re-organized our figures as recommended. We have also added an additional summary (Figure 1) to illustrate how we designated CRMs. This also addressed Reviewer 1’s concern about the lack of detail and presentation of the CRM analysis. We have also condensed our functional enrichments into one Figure (see comment below) and condensed the section on the HGMD enrichments. Overall we feel these changes have strengthened the paper and we thank the reviewer for this suggestion.

*2)*
Figure 2
*and the many supplemental figures have multiple problems. The authors have developed an excellent way of displaying the GREAT results, but they flood the reader with too much information. A better-digested and focused set of key results needs to be featured. The text does have a good focus; however, the text and figures are not closely aligned. The specific examples of significant enrichments in the text are not listed in the main*
Figure 2*, and the reader has to scan through four or five pages of figure supplements to find the data being discussed. One figure highlighting the key enrichments needs to be refined from all this material. Also, the method for assigning likely target genes is not stated in the main text. This is a very important point for interpreting the data, and it should be stated in the Results as well as in the Methods*.

We have now condensed the enrichment analysis into one Figure (Figure 3) as suggested. This allowed us to remove the many pages of supplemental Figure 2. We also used this to condense the Figure 5 results. We achieved this by recording the top 5 enrichments for the three select databases we discussed in the main text. We then plot all enrichments for any database ontology that is significant in one or more of the comparisons. This should allow the reader to more easily cross-compare the enrichments obtained for different phylogenetic categories, CRMs versus singletons and TF-specific enrichments obtained. The full list of enrichments, associated genes and statistics remain in [Supplementary-material SD8-data].

We also state in the Results: “We used the enrichment tool GREAT (48) to perform functional enrichments. GREAT’s default setting assigns TF binding events to a basal region around every gene (5 kb upstream, 1 kb downstream). ChIP-seq peaks that fall within the basal regulatory region of each gene, as well as the genomic sequence that spans between the basal region of that gene and the nearest gene’s basal region (within a maximum of 1 Mb) are used to generate functional enrichments. The most significant enrichments that were unique to the shared liver CRMs include: liver disease (Binomial FDR q-value = 4.53 x 10^-130^) from the Disease Ontology database and metabolism of lipids and lipoproteins (q = 2.96 x 10^-73^) from MSigDB Pathway (Figure 3; [Supplementary-material SD3-data]).”

*3) The description and interpretation of the data in*
Figure 4
*need to be more accurate. The data in panel B are over-interpreted in the legend and in the Results. The expression of presumptive targets is not limited to liver (“uniquely expressed”) but the expression levels are higher there. The text says “Unlike CRMs, conserved singletons do not show a strong increase in liver-specific gene expression.” In fact, presumptive targets of singletons show a similar pattern to the presumptive targets of CRMs, but the levels of expression are lower*.

We revised the description in the legend for Figure 4 accordingly: “In contrast, gene expression levels near TFs in shared CRMs were significantly higher than those near shared singleton-associated genes (p=2.4 x 10^-16^) (Figure 5). Using a reference transcription dataset that comprises RNA-seq data for liver and 15 additional human tissue types (E-MTAB-513) we confirmed the above observation and found that the gene expression association with liver CRMs, and to a lesser extent Singletons, was tissue-specific (Figure 5). In sum, deeply shared CRMs are associated with genes that are highly expressed in a distinctly liver-specific manner.”

*Is the decrease in signal for deeply shared singletons significant, or is this related to a small number of targets to examine? Also, the legend should specify the RNA-seq data used in panel A. The top comparison in*
Figure 4
*seems to be between the expression levels of presumptive targets for deeply conserved CRMs and lineage-specific singletons, but it has a surprisingly high p-value; this should be clarified*.

We have clarified the data source for Figure 4 panel A (now Figure 5) and provided more details on the analysis in the Methods section. We have added the number of gene associations for Figure 5 in the legend and 5B on the heatmap itself. To clarify the source of the high p-value the 5-way shared CRMs versus singletons, we also re-plotted the figure for each TF separately, along with the number of TF-associated genes (Figure 5—figure supplement 1). The p-values are less striking but still significant when broken down into individual factors (p<10^-3^). The reason for this high p-value is likely due to the fact that some very highly expressed liver genes have multiple CRMs close by. Our plot does not show the outliers, which include highly expressed liver genes like Albumin, fibrinogen (FGA, FGG, FGB), complement genes, and many of the acute phase response genes (e.g. CRP, SAA1, SAA2). This analysis has been added to the Results section.

*4)*
Figure 6
*main and Figure 6–figure supplement 1: The authors are making a critical point but the connections between the text and the Figures are not stated clearly. In the Results, the authors give p-values for enrichment of about 0.005 to 0.008 and refer to*
Figure 6*. However, those p-values are MUCH smaller, and correspond to the analysis in the next sentence, after separation by conservation categories. However, the reader is referred to Supplemental File 4, whereas they really need to look at*
Figure 6*! The fact that the enrichments are robust to different ways of dealing with potentially functional SNPs in proximity or LD is a strong point and accurately stated in the Results. However, the titles to the figure legends (*Figure 6
*vs. Figure 6–figure supplement 1) are the same, and it takes some careful reading to figure out what is different about the two figures*.

We have now removed the results for the “Any shared” enrichments from the text (they are still in the figure and in the supplemental file), and focused our discussion on the deeply shared CRMs. We have corrected the Figure legends. We have also redone Table 2 (now Table 1) so it includes more interesting information regarding the gene loci and disease trait categories that were enriched for deeply shared CRMs.

*5) The Discussion should address what is missed by focusing on only combinatorial binding, only conserved binding, and both. Clearly, many fewer bound sites are examined by doing this filtering, but the enrichment for likely function is substantial*. *What are the potential costs of taking this approach?*

We have now included this important point in the Discussion, with the text starting “Just as using DNA constraint as a sole criterion for enhancer finding has its limitations, using filtering approaches that require conserved combinatorial binding will also miss important regulatory events near genes of interest”.

*6) I would like to see a paragraph in the Discussion comparing the advantages of filtering vs integration*. *Are there integrative approaches that would emphasize the features shown to be useful here (combinatorial binding and conservation) while not completely excluding other binding sites?*

We have addressed this comment in the Reviewer 1, comment 1 and have created a paragraph in the Discussion, which should address comment 5 above as well.

*7) What is the null model for the tests of enrichment for GWAS SNPs in CRMs, singletons (*Figure 6*)? In previous studies, the null model was restricted to SNPs on the genotyping array that were not associated with a trait but were matched to the trait-associated SNPs in several important ways, e.g. allele frequency, position relative to gene structure, and proximity to the transcription start site*.

We implemented two null models to further test the significance of the enrichments we found. First we shuffled the GWAS SNP summary annotation 1,000 times. Using each random set, we annotated CRMs and singletons with the summaries assigned to overlapping SNPs as described for Figure 7. We then calculated the enrichment for each summary term in the various phylogenetic categories. Second, we created a random set of SNPs from the full list of GWAS lead SNPs that matched the minor allele frequency and distance to transcription start site that we observed in our analyses in Figure 7. This was repeated 1,000 times to create a null distribution of p-values. We considered an enrichment significant if less than 5% of the 1,000 p-values obtained from either of the two null models had an enrichment p-value equal or lower to the one obtained using the original GWAS SNPs summary annotation sample before multiple testing correction.